# PROVABLY ROBUST ADVERSARIAL EXAMPLES

**Dimitar I. Dimitrov**[1]     **Gagandeep Singh**[2,3]     **Timon Gehr**[1]     **Martin Vechev**[1]

[1] ETH Zurich    [2] University of Illinois, Urbana Champaign    [3] VMware Research
`{dimitar.dimitrov, timon.gehr, martin.vechev}@inf.ethz.ch`[1]
`ggnds@illinois.edu`[2]

## ABSTRACT

We introduce the concept of provably robust adversarial examples for deep neural networks – connected input regions constructed from standard adversarial examples which are *guaranteed* to be robust to a set of real-world perturbations (such as changes in pixel intensity and geometric transformations). We present a novel method called PARADE for generating these regions in a scalable manner which works by iteratively refining the region initially obtained via sampling until a refined region is certified to be adversarial with existing state-of-the-art verifiers. At each step, a novel optimization procedure is applied to maximize the region's volume under the constraint that the convex relaxation of the network behavior with respect to the region implies a chosen bound on the certification objective. Our experimental evaluation shows the effectiveness of PARADE: it successfully finds large provably robust regions including ones containing $\approx 10^{573}$ adversarial examples for pixel intensity and $\approx 10^{599}$ for geometric perturbations. The provability enables our robust examples to be significantly more effective against state-of-the-art defenses based on randomized smoothing than the individual attacks used to construct the regions.

## 1 INTRODUCTION

Deep neural networks (DNNs) are vulnerable to adversarial attacks: small input perturbations that cause misclassification (Szegedy et al., 2013). This has caused an increased interest in investigating powerful attacks (Goodfellow et al., 2015; Carlini & Wagner, 2017; Madry et al., 2018; Andriushchenko et al., 2019; Zheng et al., 2019; Wang et al., 2019; Croce & Hein, 2019; Tramèr et al., 2020). An important limitation of existing attack methods is that they only produce a single *concrete* adversarial example and their effect can often be mitigated with existing defenses (Madry et al., 2018; Hu et al., 2019; Sen et al., 2020; Xiao et al., 2020; Pang et al., 2020; Lécuyer et al., 2019; Cohen et al., 2019; Salman et al., 2019; Fischer et al., 2020; Li et al., 2020). The effectiveness of these attacks can be improved by *robustifying* them: to consolidate the individual examples to produce a large symbolic region guaranteed to only contain adversarial examples.

**This Work: Provably Robust Adversarial Examples.** We present the concept of a *provably robust adversarial example* – a tuple consisting of an adversarial input point and an input region around it capturing a very large set (e.g., $> 10^{100}$) of points guaranteed to be adversarial. We then introduce a novel algorithm for generating such regions and apply it to the setting of pixel intensity changes and geometric transformations. Our work relates to prior approaches on generating robust adversarial examples (Athalye et al., 2018; Qin et al., 2019) but differs in a crucial point: our regions are *guaranteed* to be adversarial while prior approaches are empirical and offer no such guarantees.

**Main Contributions.** Our key contributions are:

- The concept of a *provably robust adversarial example* – a connected input region capturing a very large set of points, generated by a set of perturbations, guaranteed to be adversarial.

- A novel scalable method for synthesizing such examples called ProvAbly Robust ADversarial Examples (PARADE), based on iterative refinement that employs existing state-of-the-art techniques to certify the robustness. Our method is compatible with a wide range of certifi-

cation techniques making it easily extendable to new adversarial attack models. We make the code of PARADE available at `https://github.com/eth-sri/parade.git`

- A thorough evaluation of PARADE, demonstrating it can generate provable regions containing $\approx 10^{573}$ concrete adversarial points for pixel intensity changes, in $\approx 2$ minutes, and $\approx 10^{599}$ concrete points for geometric transformations, in $\approx 20$ minutes, on a challenging CIFAR10 network. We also demonstrate that our robust adversarial examples are significantly more effective against state-of-the-art defenses based on randomized smoothing than the individual attacks used to construct the regions.

## 2 BACKGROUND

We now discuss the background necessary for the remainder of the paper. We consider a neural network $f \colon \mathbb{R}^{n_0} \to \mathbb{R}^{n_l}$ with $l$ layers, $n_0$ input neurons and $n_l$ output classes. While our method can handle arbitrary activations, we focus on networks with the widely-used ReLU activation. The network classifies an input $x$ to class $y(x)$ with the largest corresponding output value, i.e., $y(x) = \operatorname{argmax}_i [f(x)]_i$. Note for brevity we omit the argument to $y$ when it is clear from the context.

### 2.1 NEURAL NETWORK CERTIFICATION

In this work, we rely on existing state-of-the-art neural network certification methods based on convex relaxations to prove that the adversarial examples produced by our algorithm are robust. These certification methods take a convex input region $\mathcal{I} \subset \mathbb{R}^{n_0}$ and prove that every point in $\mathcal{I}$ is classified as the target label $y_t$ by $f$. They propagate the set $\mathcal{I}$ through the layers of the network, producing a convex region that covers all possible values of the output neurons (Gehr et al., 2018). Robustness follows by proving that, for all combinations of output neuron values in this region, the output neuron corresponding to class $y_t$ has a larger value than the one corresponding to any other class $y \neq y_t$.

Commonly, one proves this property by computing a function $L_y \colon \mathbb{R}^{n_0} \to \mathbb{R}$ for each label $y \neq y_t$, such that, for all $x \in \mathcal{I}$, we have $L_y(x) \leq [f(x)]_{y_t} - [f(x)]_y$. For each $L_y$, one computes $\min_{x \in \mathcal{I}} L_y(x)$ to obtain a global lower bound that is true for all $x \in \mathcal{I}$. If we obtain positive bounds for all $y \neq y_t$, robustness is proven. To simplify notation, we will say that the *certification objective* $L(x)$ is the function $L_y(x)$ with the smallest minimum value on $\mathcal{I}$. We will call its corresponding minimum value the *certification error*. We require $L_y(x)$ to be a linear function of $x$. This requirement is consistent with many popular certification algorithms based on convex relaxation, such as CROWN (Zhang et al., 2018), DeepZ (Singh et al., 2018a), and DeepPoly (Singh et al., 2019). Without loss of generality, for the rest of this paper, we will treat DeepPoly as our preferred certification method.

### 2.2 CERTIFICATION AGAINST GEOMETRIC TRANSFORMATIONS

DeepPoly operates over specifications based on linear constraints over input pixels for verification. These constraints are straightforward to provide for simple pixel intensity transformations such as adversarial patches (Chiang et al., 2020) and $L_\infty$ (Carlini & Wagner, 2017) perturbations that provide a closed-form formula for the input region. However, geometric transformations do not yield such linear regions. To prove the robustness of our generated examples to geometric transformations, we rely on DeepG (Balunović et al., 2019) which, given a range of geometric transformation parameters, creates an overapproximation of the set of input images generated by the geometric perturbations. DeepG then leverages DeepPoly to certify the input image region. When generating our geometric robust examples, we work directly in the geometric parameter space and, thus, our input region $\mathcal{I}$ and the inputs to our certification objective $L(x)$ are also in geometric space. Despite this change, as our approach is agnostic to the choice of the verifier, in the remainder of the paper we will assume the certification is done using DeepPoly and not DeepG, unless otherwise stated.

### 2.3 RANDOMIZED SMOOTHING

Randomized smoothing (Lécuyer et al., 2019; Cohen et al., 2019) is a provable defense mechanism against adversarial attacks. For a chosen standard deviation $\sigma$ and neural network $f$ as defined above, randomized smoothing computes a smoothed classifier $g$ based on $f$, such that $g(x) = \operatorname{argmax}_c \mathbb{P}(y(x + \epsilon) = c)$ with random Gaussian noise $\epsilon \sim \mathcal{N}(0, \sigma^2 I)$. This construction of $g$ allows

Cohen et al. (2019) to introduce the procedure CERTIFY that provides probabilistic guarantees on the robustness of $g$ around a point $x$:

**Proposition 1.** *(From Cohen et al. (2019)) With probability at least $1 - \alpha$ over the randomness in* CERTIFY, *if* CERTIFY *returns a class $y$ and a radius R (i.e does not abstain), then g predicts y within radius R around $x$ : $g(x + \delta) = y$, for all $\|\delta\|_2 < R$.*

We define adversarial attacks on smoothed classifiers, as follows:

**Definition 1** (Adversarial attack on smoothed classifiers). *For a fixed $\sigma$, $\alpha$ and an adversarial distance $R' \in \mathbb{R}^{>0}$, we call $\tilde{x} \in \mathbb{R}^{n_0}$ an adversarial attack on the smoothed classifier $g$ at the point $x \in \mathbb{R}^{n_0}$, if $\|\tilde{x} - x\|_2 < R'$ and $g(x) \neq g(\tilde{x})$.*

Similarly to generating adversarial attacks on the network $f$, we need to balance the adversarial distance $R'$ on $g$. If too big — the problem becomes trivial; if too small — no attacks exist. We outline the exact procedure we use to heuristically select $R'$ in Appendix C.4. Using the above definition, we define the strength of an attack $\tilde{x}$ as follows:

**Definition 2** (Strength of adversarial attack on smoothed classifiers). *We measure the strength of an attack $\tilde{x}$ in terms of $R_{adv}$ – the radius around $\tilde{x}$, whose $L_2$ ball is certified to be the same adversarial class as $\tilde{x}$ on the smoothed network $g$ using* CERTIFY *for a chosen $\sigma$ and $\alpha$.*

Intuitively, this definition states that for points $\tilde{x}$ for which $R_{adv}$ is bigger, the smoothed classifier is less confident about predicting the correct class, since more adversarial examples are sampled in this region and therefore, the attack on $g$ is stronger. We use this measure in Section 5 to compare the effectiveness of our adversarial examples to examples obtained by PGD on $g$.

## 3 OVERVIEW

Existing methods for generating robust adversarial examples focus on achieving empirical robustness (Qin et al., 2019; Athalye et al., 2018). In contrast, we consider *provably* robust adversarial examples, defined below:

**Definition 3** (Provably Robust Adversarial Example). *We define a provably robust adversarial example to be any large connected neural network input region, defined by a set of perturbations of an input, that can be formally proven to only contain adversarial examples.*

In this section, we outline how PARADE generates such regions. The technical details are given in Section 4. To generate a provably robust adversarial example, ideally, we would like to directly maximize the input region's size, under the constraint that it only contains adversarial examples. This leads to multiple challenges: Small changes of the parametrization of the input region (e.g., as a hyperbox) can lead to large changes of the certification objective, necessitating a small learning rate for optimization algorithms based on gradient descent. At the same time, the optimizer would have to solve a full forward verification problem in each optimization step, which is slow and impractical. Additionally, like Balunovic & Vechev (2020), we empirically observed that bad initialization of the robust region causes convergence to local minima, resulting in small regions. This can be explained by the practical observation in Jovanović et al. (2021), which shows that optimization problems involving convex relaxations are hard to solve for all but the simplest convex shapes. We now provide an overview of how we generate robust adversarial regions while alleviating these problems.

Our method for generating robust examples in the shape of a hyperbox is shown in Figure 1 and assumes an algorithm $\mathbb{A}$ that generates adversarial examples and a neural network verifier $\mathbb{V}$. We require $\mathbb{V}$ to provide the certification objective $L(x)$ as a linear function of the network's input neurons (whose values can be drawn from the input hyperbox), as described in Section 2. We split our algorithm into two steps, described in Section 3.1 and Section 3.2, to address the challenges outlined above. An optional final step, illustrated in Figure 2 and outlined in Appendix E, demonstrates the extension of PARADE to generate polyhedral examples.

### 3.1 COMPUTING AN OVERAPPROXIMATION REGION

In the first step, we compute an overapproximation hyperbox $\mathcal{O}$ by fitting a hyperbox around samples obtained from $\mathbb{A}$. $\mathcal{O}$ is represented as a dashed blue rectangle in Figure 1. Intuitively, we use $\mathcal{O}$ to restrict the search space for adversarial regions to a part of the input space that $\mathbb{A}$ can attack.

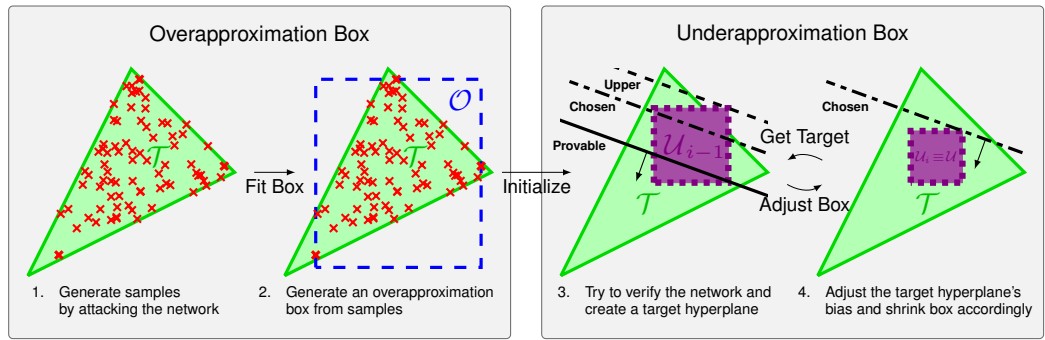

Figure 1: Overview of PARADE. The green triangle denoted $\mathcal{T}$ represents the ground truth adversarial region and the red crosses represent the attacks obtained by $\mathbb{A}$. The dashed blue and violet dotted rectangles denoted $\mathcal{O}$ and $\mathcal{U}$, represent the fitted overapproximation and underapproximation boxes, respectively. The solid black lines represent hyperplanes generated by the verification procedure, while their dash-dotted counterparts represent the hyperplanes after bias-adjustment. The arrows going out of the hyperplanes represent the direction of their corresponding half-spaces.

## 3.2 COMPUTING AN UNDERAPPROXIMATION REGION

In the second step, we compute the robust adversarial example as a hyperbox $\mathcal{U}$, represented as a dotted violet rectangle in Figure 1. We will refer to $\mathcal{U}$ as underapproximation hyperbox to distinguish it from the polyhedral robust example $\mathcal{P}$ generated in Appendix E. $\mathcal{U}$ is obtained by iteratively shrinking $\mathcal{O}$ until arriving at a provably adversarial region. We denote the hyperbox created at iteration $i$ of the shrinking process with $\mathcal{U}_i$, where $\mathcal{U}_0 = \mathcal{O}$. At each iteration $i$, we compute the certification objective $L^{i-1}(x)$ of the region $\mathcal{U}_{i-1}$ computed in the previous iteration. If the objective is proven to be positive, we assign $\mathcal{U}$ to $\mathcal{U}_{i-1}$, as in Step 4 in Figure 1, and return. Otherwise, we solve an optimization problem, optimizing a surrogate for the volume of $\mathcal{U}_i \subseteq \mathcal{U}_{i-1}$ under the constraint that $\min_{x \in \mathcal{U}_i} L^{i-1}(x) \geq -p_{i-1}$, for a chosen bound $p_{i-1} \geq 0$. We describe the optimization problem involved in the shrinking procedure in the next paragraph. In Section 4, we will present and motivate our choice of $p_{i-1}$. In Appendix D.2, we further discuss a sufficient condition under which the overall procedure for generating $\mathcal{U}$ is guaranteed to converge and why this condition often holds in practice.

An important property of our choice of $p_{i-1}$ is that it becomes smaller over time and thus forces the certification objective to increase over time until it becomes positive. We note that optimizing $L(x)$ this way is simpler than directly optimizing for $\mathcal{U}$ by differentiating through the convex relaxation of the network. In particular, our method does not require propagating a gradient through the convex relaxation of the network and the convex relaxation is updated only once per iteration. Additionally, it allows PARADE to work with convex relaxation procedures that are not differentiable, such as the DeepG method employed in this paper. The resulting hyperbox $\mathcal{U}$ is a provable adversarial region.

**Shrinking $\mathcal{U}_{i-1}$.** There are multiple ways to shrink $\mathcal{U}_{i-1}$ so that the condition $\min_{x \in \mathcal{U}_i} L^{i-1}(x) \geq -p_{i-1}$ is satisfied. We make the greedy choice to shrink $\mathcal{U}_{i-1}$ in a way that maximizes a proxy for the volume of the new box $\mathcal{U}_i$. We note this may not be the globally optimal choice across all iterations of the algorithm, however, and that even achieving local optimality in terms of volume of $\mathcal{U}_i$ is a hard problem as it is non-convex in high dimensions.

One possible solution to the shrinking problem is to use a scheme that we call uniform shrinking, where all lower bounds $[l^{i-1}]_j$ and upper bounds $[u^{i-1}]_j$ in each dimension $j$ of the hyperbox $\mathcal{U}_{i-1}$ are changed by the same amount. The optimal amount for this uniform shrinking scheme can then be selected using a binary search. While this gives an efficient shrinking algorithm, in practice we observe that uniform shrinking produces small regions. This is expected, as it may force all dimensions of the hyperbox to be shrunk by a large amount even though it might be enough to shrink only along some subset of the dimensions.

In contrast, our shrinking scheme adjusts each $[l^{i-1}]_j$ and $[u^{i-1}]_j$ individually. While this shrinking scheme is more powerful, optimizing the size of $\mathcal{U}_i$ with respect to its lower and upper bounds $[l^i]_j$ and $[u^i]_j$ becomes more complex. We approach the problem differently in the high dimensional input setting related to $L_\infty$ perturbations and the low dimensional setting related to geometric changes.

**Shrinking for $L_\infty$ Transformations.** For the high-dimensional setting of $L_\infty$ pixel transformations, we maximize $\sum_{j=1}^{n_0} [w^i]_j$, where $[w^i]_j = [u^i]_j - [l^i]_j$ denotes the width of the $j^{\text{th}}$ dimension of $\mathcal{U}_i$. The linear objective allows for solutions where a small number of dimensions are shrunk to 0 width

in favor of allowing most input dimensions to retain a big width. As a result, in our computed region $\mathcal{U}_i$ there are few pixels that can only take a single value while most other pixels can take a big range of values. Since, $\sum_{i=j}^{n_0} [w^i]_j$ is a linear function of $[l^{i-1}]_j$ and $[u^{i-1}]_j$, the optimization can be posed as a Linear Program (LP) (Schrijver, 1998) and be efficiently solved with an LP solver to obtain optimal values for $[u^i]_j$ and $[l^i]_j$ w.r.t the objective, as we show in Appendix B.2.

**Shrinking for Geometric Transformations.** As discussed in Section 2.2, we generate our provably robust examples to geometric transformations in the low-dimensional geometric parameter space. Therefore, the resulting region $\mathcal{U}_i$ is also low-dimensional. We found that maximizing the logarithm of the volume of $\mathcal{U}_i$ is more effective in this setting, as this objective encourages roughly equally-sized ranges for the dimensions of $\mathcal{U}_i$. We solve the optimization problem using PGD, which can lead to merely locally-optimal solutions for the volume of $\mathcal{U}_i$. We, therefore, rely on multiple initializations of PGD and choose the best result. Appendix B.3 describes the process in detail.

### 3.3 PROVABLY ROBUST POLYHEDRAL ADVERSARIAL EXAMPLES

We remark that our provably robust adversarial examples are not restricted to hyperbox shapes. Appendix E describes an optional final step of PARADE that generates provably robust polyhedral adversarial examples $\mathcal{P}$ from $\mathcal{U}$ and $\mathcal{O}$. $\mathcal{P}$ is iteratively cut from $\mathcal{O}$, such that $\mathcal{P} \supseteq \mathcal{U}$ is ensured. Therefore, $\mathcal{P}$ represents a bigger region, but takes additional time to compute. The computation of $\mathcal{P}$, requires a $\mathbb{V}$ that accumulates imprecision only at activation functions, such as ReLU or sigmoid, in order to work. This property is violated by DeepG's handling of the geometric transformations. For this reason, we only apply it in the $L_\infty$-transformation setting. We refer interested readers to Appendix E for information on how we generate polyhedral adversarial examples.

## 4 PARADE: PROVABLY ROBUST ADVERSARIAL EXAMPLES

We now present PARADE in more formal terms. We already discussed how to compute $\mathcal{O}$ in Section 3. We use Algorithm 1 to compute $\mathcal{U}$, which is the output of PARADE. It requires a neural network $f \colon \mathbb{R}^{n_0} \to \mathbb{R}^{n_l}$ with $l$ layers, adversarial target class $y_t$, an overapproximation hyperbox $\mathcal{O}$, a speed/precision trade-off parameter $c \in [0, 1]$, a maximum number of iterations $u_\text{it}$, an early termination threshold $t$ and a certification method $\mathbb{V}$ that takes a neural network $f$, a hyperbox $\mathcal{U}_{i-1}$ and a target $y_t$ and returns a linear certification objective $L^{i-1}(x)$.

---

**Algorithm 1** GENERATE_UNDERAPPROX

1: **func** GENERATE_UNDERAPPROX( $f$, $\mathcal{O}$, $\mathbb{V}$, $y_t$, $c$, $u_\text{it}$, $t$ )
2:     $\mathcal{U}_0 = \mathcal{O}$
3:     **for** $i \in \{1, 2, \ldots, u_\text{it}\}$ **do**
4:         $L^{i-1}(x), e^{i-1} = \mathbb{V}(f, \mathcal{U}_{i-1}, y_t)$
5:         **if** $e^{i-1} \geq 0$ **then**
6:             **return** $\mathcal{U}_{i-1}$
7:         **end if**
8:         $p_{i-1} = -e^{i-1} \cdot c$
9:         **if** $p_{i-1} \leq t$ **then**
10:            $p_{i-1} = 0$
11:         **end if**
12:         **if** $L_\infty\_transformation$ **then**
13:            $\mathcal{U}_i = \text{Shrink\_LP}(\mathcal{U}_{i-1}, L^{i-1}(x), p_{i-1})$
14:         **else**
15:            $\mathcal{U}_i = \text{Shrink\_PGD}(\mathcal{U}_{i-1}, L^{i-1}(x), p_{i-1})$
16:         **end if**
17:     **end for**
18:     **return** FailedToConverge
19: **end func**

---

As described in Section 3.2, the algorithm generates a sequence of hyperboxes $\mathcal{U}_0 \supseteq \mathcal{U}_1 \supseteq \ldots \supseteq \mathcal{U}_{u_\text{it}}$ with $\mathcal{U}_0 = \mathcal{O}$ and returns the first hyperbox from the sequence proven to be adversarial by $\mathbb{V}$. At each iteration, the algorithm attempts to certify the hyperbox from the previous iteration $\mathcal{U}_{i-1}$ to be adversarial by computing the certification error $e^{i-1}$ (Line 4) and checking if it is positive (Line 5). If successful, $\mathcal{U}_{i-1}$ is returned. Otherwise, we use $L^{i-1}(x)$ generated by our attempt at verifying $\mathcal{U}_{i-1}$ (Line 4) and generate the constraint $\min_{x \in \mathcal{U}_i} L^{i-1}(x) \geq -p_{i-1}$ based on a parameter $p_{i-1} \geq 0$. We note that $\min_{x \in \mathcal{U}_i} L^{i-1}(x)$ is a function of the hyperbox $\mathcal{U}_i$, parametrized by its lower and upper bounds. In order to shrink $\mathcal{U}_{i-1}$, we optimize the lower and upper bounds of $\mathcal{U}_i$ explicitly, such that the constraint holds true for the bounds. Next, we specify and motivate our choice of $p_{i-1}$.

At each iteration $i$, we use the constraint $\min_{x \in \mathcal{U}_i} L^{i-1}(x) \geq -p_{i-1}$ to shrink $\mathcal{U}_{i-1}$ (Line $12-16$) by calling either the SHRINK_LP or SHRINK_PGD optimization procedures. SHRINK_LP and

Table 1: Comparison between different methods for creating adversarial examples robust to intensity changes. Column $\epsilon$ depicts the preturbation radius used by adversarial algorithm $\mathbb{A}$. Columns #Corr, #Img, #Reg show the number of correctly classified images, adversarial images and adversarial regions for the network. For each method columns #VerReg, Time and #Size show the number of verified regions, average time taken, and number of concrete adversarial examples inside the regions.

| | | | | | | BASELINE | | | PARADE Box | | | PARADE Poly | | |
|---|---|---|---|---|---|---|---|---|---|---|---|---|---|---|
| DATASET | MODEL | $\epsilon$ | #CORR | #IMG | #REG | #VERREG | TIME | SIZE | #VERREG | TIME | SIZE | #VERREG | TIME | SIZEO |
| MNIST | 8 x 200 | 0.045 | 97 | 22 | 53 | 41 | 272 s | $10^{24}$ | 53 | 114 s | $10^{121}$ | 53 | 1556 s | $< 10^{191}$ |
| | CONVSMALL | 0.12 | 100 | 21 | 32 | 31 | 171 s | $10^{339}$ | 32 | 74 s | $10^{494}$ | 32 | 141 s | $< 10^{561}$ |
| | CONVBIG | 0.05 | 98 | 18 | 29 | 15 | 1933s | $10^{9}$ | 28 | 880 s | $10^{137}$ | 28 | 5636 s | $< 10^{173}$ |
| CIFAR10 | CONVSMALL | 0.006 | 59 | 23 | 44 | 28 | 238 s | $10^{360}$ | 44 | 113 s | $10^{486}$ | 44 | 264 s | $< 10^{543}$ |
| | CONVBIG | 0.008 | 60 | 25 | 36 | 26 | 479 s | $10^{380}$ | 36 | 404 s | $10^{573}$ | 36 | 610 s | $< 10^{654}$ |

SHRINK_PGD are detailed in Appendix B.2 and Appendix B.3, respectively. To enforce progress of the algorithm, we require the constraint to remove non-zero volume from $\mathcal{U}_{i-1}$. Let $p_{i-1}^{\max}$ be any upper bound on $p_{i-1}$, for which if $p_{i-1}$ exceeds $p_{i-1}^{\max}$ the constraint $\min_{x \in \mathcal{U}_i} L^{i-1}(x) \geq -p_{i-1}$ is trivially satisfied with $\mathcal{U}_i = \mathcal{U}_{i-1}$. We can show $p_{i-1}^{\max} = -\min_{x \in \mathcal{U}_{i-1}} L^{i-1}(x)$ is one such upper bound, as we have: $\min_{x \in \mathcal{U}_i} L^{i-1}(x) \geq \min_{x \in \mathcal{U}_{i-1}} L^{i-1}(x) = -p_{i-1}^{\max}$, where the inequality follows from $\mathcal{U}_i \subseteq \mathcal{U}_{i-1}$. We show the hyperplane $L^{i-1}(x) = -p_{i-1}^{\max}$ denoted as *Upper* in Step 3 in Figure 1. We note that *Upper* always touches $\mathcal{U}_{i-1}$ by construction. Decreasing the value of $p_{i-1}$, increases the volume removed from $\mathcal{U}_{i-1}$, as demonstrated by the other two hyperplanes in Step 3 in Figure 1. Additionally, we require $p_{i-1} \geq 0$ because any value of $p_{i-1} \leq 0$ creates a provably adversarial region $\mathcal{U}_i$ and choosing $p_{i-1} < 0$ generates smaller region $\mathcal{U}_i$ than choosing $p_{i-1} = 0$. We denote the hyperplane corresponding to $p_{i-1} = 0$ in Figure 1 as *Provable*. We, thus, use $p_{i-1} \in [0, p_{i-1}^{\max})$.

We note that while $p_{i-1} = 0$ is guaranteed to generate a provable region $\mathcal{U}_i$, it is often too greedy. This is due to the fact that with an imprecise $\mathbb{V}$, $L^{i-1}(x)$ itself is also imprecise and, therefore a more precise certification method may certify $\mathcal{U}_i$ as adversarial for $p_{i-1} > 0$. Further, as the region $\mathcal{U}_{i-1}$ shrinks, the precision of $\mathbb{V}$ increases. Thus, $\mathbb{V}$ might be capable of certifying $\mathcal{U}_i$, for $p_{i-1} > 0$, since on the next iteration of the algorithm the more precise certification objective $L^i(x)$ will be used for certifying the robustness of the smaller $\mathcal{U}_i$. We choose $p_{i-1}$ heuristically, by setting $p_{i-1} = p_{i-1}^{\max} \cdot c$, for a chosen parameter $c \in [0, 1)$. This is depicted in Line 8 in Algorithm 1. The closer $c$ is to 1, the bigger the chance that the modified region will not verify, but, also, the bigger the region we produce and vice versa. Thus, $c$ balances the precision and the speed of convergence of the algorithm. We empirically verify the effect of $c$ on our regions in Appendix A.1. The hyperplane resulting from $c = 0.65$, denoted as *Chosen*, is shown in Step 3 in Figure 1.

Finally, if at some iteration $i$, the *Upper* hyperplane gets closer than a small threshold $t$ to the *Provable* hyperplane, that is $p_{i-1} \leq t$, we set $p_{i-1} = 0$ to force the algorithm to terminate (Line 10). This speeds up the convergence of PARADE in the final stages, when the precision loss of $\mathbb{V}$ is not too big.

## 5 EXPERIMENTAL EVALUATION

We now evaluate the effectiveness of PARADE on realistic networks. We implemented PARADE in Python and used Tensorflow (Abadi et al., 2015) for generating PGD attacks. We use Gurobi 9.0 (Gurobi Optimization, LLC, 2020) for solving the LP instances. We rely on ERAN (Singh et al., 2018b) for its DeepPoly and DeepG implementations. We ran all our experiments on a 2.8 GHz 16 core Intel(R) Xeon(R) Gold 6242 processor with 64 GB RAM.

**Neural Networks.** We use MNIST (LeCun et al., 1998) and CIFAR10 (Krizhevsky, 2009) based neural networks. Table 5 in Appendix C.1 shows the sizes and types of the different networks. All networks in our $L_\infty$ experiments, except the CIFAR10 `ConvBig`, are not adversarially trained. In our experiments, normally trained networks were more sensitive to geometric perturbations than for $L_\infty$. Therefore, we use DiffAI-trained (Mirman et al., 2018) networks which tend to be less sensitive against geometric transformations. Our deepest network is the MNIST $8 \times 200$ fully-connected network (FFN) which has 8 layers with 200 neurons each and another layer with 10 neurons. Our largest network is the CIFAR10 `ConvBig` network with 6 layers and $\approx 62$K neurons. This network is among the largest benchmarks that existing certification methods can handle in a scalable and precise manner. We compare the methods on the first 100 test images of the datasets, a standard practice in the certification literature (Balunović et al., 2019; Singh et al., 2019), while filtering the wrongly-classified ones. For each experiment, we tuned the value of $c$ to balance runtime and size.

Table 2: Comparison between methods for creating adversarial examples robust to geometric changes. Columns #Corr, #Img, #Reg show the number of correctly classified images, adversarial images and adversarial regions for the network. For each method columns #VerReg, Time and #Splits show the number of verified regions, average time taken, and number of splits used to verify. Under and Over show median bounds on the number of concrete adversarial examples inside the regions.

| | | | | | BASELINE | | | PARADE | | | | |
|---|---|---|---|---|---|---|---|---|---|---|---|---|
| DATASET | TRANSFORM | #CORR | #IMG | #REG | #VERREG | TIME | #SPLITS | #VERREG | TIME | #SPLITS | UNDER | OVER |
| MNIST CONVSMALL | R(17) Sc(18) Sh(0.03) | 99 | 38 | 54 | 10 | 890 s | 2x5x2 | 51 | 774 s | 1x2x1 | $> 10^{96}$ | $< 10^{195}$ |
| | Sc(20) T(-1.7,1.7,-1.7,1.7) | 99 | 32 | 56 | 5 | 682 s | 4x3x3 | 51 | 521 s | 2x1x1 | $> 10^{71}$ | $< 10^{160}$ |
| | Sc(20) R(13) B(10, 0.05) | 99 | 33 | 48 | 2 | 420 s | 3x2x2x2 | 40 | 370 s | 2x1x1x1 | $> 10^{70}$ | $< 10^{455}$ |
| MNIST CONVBIG | R(10) Sc(15) Sh(0.03) | 95 | 40 | 50 | 9 | 812 s | 2x4x2 | 44 | 835 s | 1x2x1 | $> 10^{77}$ | $< 10^{205}$ |
| | Sc(20) T(0,1,0,1) | 95 | 34 | 46 | 2 | 435 s | 4x2x2 | 42 | 441 s | 2x1x1 | $> 10^{64}$ | $< 10^{174}$ |
| | Sc(15) R(9) B(5, 0.05) | 95 | 39 | 52 | 2 | 801 s | 3x2x2x2 | 46 | 537 s | 2x1x1x1 | $> 10^{119}$ | $< 10^{545}$ |
| CIFAR CONVSMALL | R(2.5) Sc(10) Sh(0.02) | 53 | 24 | 29 | 1 | 1829 s | 5x2x2 | 29 | 1369 s | 2x1x1 | $> 10^{599}$ | $< 10^{1173}$ |
| | Sc(10) T(0,1,0,1) | 53 | 28 | 32 | 1 | 1489 s | 4x3x3 | 32 | 954 s | 2x1x1 | $> 10^{66}$ | $< 10^{174}$ |
| | Sc(5) R(8) B(1, 0.01) | 53 | 21 | 25 | 1 | 2189 s | 5x2x2x2 | 21 | 1481 s | 2x1x1x1 | $> 10^{513}$ | $< 10^{2187}$ |

## 5.1 ADVERSARIAL EXAMPLES ROBUST TO INTENSITY CHANGES

Table 1 summarizes our results on generating examples robust to intensity changes, whose precise definition is in Appendix B.1. Further, Appendix F.2 shows images of the examples obtained in our experiment. We compare the uniform shrinking described in Section 3, used as a baseline, against the PARADE variants for generating robust hyperbox and polyhedral adversarial examples. In all experiments, we compute $\mathcal{O}$ using examples collected within an $L_\infty$ ball around a test image with the radius $\epsilon$ specified in Table 1. The values of $\epsilon$ are chosen such that the attack has a non-trivial probability ($> 0$ but $< 1$) of success. Additional details about the experimental setup are given in Appendix C.2.

For the hyperbox adversarial examples obtained by both PARADE and the baseline, we calculate the size of the example in terms of the number of concrete adversarial examples it contains and we report the median of all regions under the Size columns in Table 1. For the polyhedral adversarial examples, we report the median number of concrete adversarial examples contained within an overapproximated hyperbox around the polyhedral region in the SizeO column. We note that our hyperbox regions are contained within our polyhedral regions by construction and, thus, the size of the hyperbox regions acts as an underapproximation of the size of our polyhedral regions. We also report the average runtime of all algorithms on the attackable regions.

We note that for MNIST `ConvBig` and $8 \times 200$, the regions obtained for $\mathcal{O}$ produced huge certification error $> 1000$, which prevented our underapproximation algorithm $\mathcal{U}$ to converge on these networks. To alleviate the problem, we used uniform shrinking to lower the error to under 100 first. We then used the obtained box in place of $\mathcal{O}$. We conjecture the reason for this issue is the huge amount of imprecision accumulated in the networks due to their depth and width and note that the uniform shrinking on these networks alone performs worse. For each network, in Table 1, we report the number of distinct pairs of an image and an adversarial target on which $\mathbb{A}$ succeeds (column #Reg), as well as the number of unique images on which $\mathbb{A}$ succeeds (column #Img). We note that #Reg > #Img. We further report the number of test images that are correctly classified by the networks (column #Corr). The table details the number of regions on which the different methods succeed (column #VerReg), that is, they find a robust adversarial example containing $> 1000$ concrete adversarial examples. As can be seen in Table 1, PARADE Box creates symbolic regions that consistently contain more concrete adversarial examples (up to $10^{573}$ images on CIFAR10 `ConvBig`) than the baseline while being faster. PARADE also succeeds more often, failing only for a single region due to the corresponding hyperbox $\mathcal{O}$ also containing $< 1000$ concrete examples versus 53 failures for the baseline. Overall, our method, unlike the baselines, generates robust examples containing large number of examples in almost all cases in which the image is attackable, in a few minutes.

## 5.2 ADVERSARIAL EXAMPLES ROBUST TO GEOMETRIC CHANGES

Next, we demonstrate the results of our algorithm for generating examples robust to geometric transformations, whose precise definition is given in Appendix B.1. Here, PARADE relies on DeepG for robustness certification, as detailed in Section 2. To increase its precision, DeepG selectively employs input space splitting, where each split is verified individually. This strategy is effective due to the low dimensionality of the input space (3 to 4 geometric parameters in our experiments). To this end, we uniformly split our method's initial region $\mathcal{O}$, instead, where all splits are individually shrunken and their sizes are summed. We compare PARADE against a baseline based on a more aggressive uniform splitting of $\mathcal{O}$ that does not use our shrinking procedure. Since the cost and

Table 3: The robustness of different examples to $L_2$ smoothing defenses.

| | MNIST | | | CIFAR | |
|---|---|---|---|---|---|
| METHOD | 8x200 | CONVSMALL | CONVBIG | CONVSMALL | CONVBIG |
| BASELINE | 0.55 | 0.38 | 0.59 | 0.53 | 0.26 |
| PARADE | **1.00** | **1.00** | **1.00** | **1.00** | **1.00** |
| INDIVIDUAL ATTACKS MEAN | 0.29 | 0.16 | 0.18 | 0.48 | 0.25 |
| INDIVIDUAL ATTACKS 95% PERCENTILE | 0.53 | 0.44 | 0.51 | 0.61 | 0.37 |

precision of verification increases with the number of splits, to allow for fair comparison we select the number of baseline splits, so the time taken by the baseline is similar or more than for PARADE.

Table 2 summarizes the results for PARADE and the baseline on the MNIST and CIFAR10 datasets. Further, Appendix F.3 shows visualizations of the adversarial examples obtained in the experiment. The Transform column in Table 2 represents the set of common geometric transformations on the images. Here, $R(x)$ signifies a rotation of up to $\pm x$ degrees; $Sc(x)$, scaling of up to $\pm x\%$; $Sh(m)$, shearing with a factor up to $\pm m\%$; $B(\gamma, \beta)$, changes in contrast between $\pm\gamma\%$ and brightness between $\pm\beta\%$; and $T(x_l, x_r, y_d, y_u)$, translation of up to $x_l$ pixels left, $x_r$ pixels right, $y_d$ pixels down and $y_u$ pixels up. The selected combinations of transformations contain $\leq 4$ parameters, as is standard for assessing adversarial robustness against geometrical perturbations (Balunović et al., 2019). Unlike DeepG, we chose only combinations of $\geq 3$ parameters, as they are more challenging. The chosen combinations ensure that all transformations from DeepG are present. Similar to $L_\infty$, we chose the parameter bounds such that the attack algorithm has a non-trivial probability of success. Additional details about the experimental setup are given in Appendix C.3.

We note that in this setting PARADE creates robust regions in the geometric parameter space, however since we are interested in the number of concrete adversarial images possible within our computed regions, we use DeepG to compute a polyhedron overapproximating the values of the pixels inside the region. For each polyhedron, we compute underapproximation and overapproximation hyperboxes, resulting in a lower and an upper bound on the number of concrete examples within our computed regions. The underapproximation hyperbox is computed as in Gopinath et al. (2019) and the overapproximation is constructed by computing the extreme values of each pixel. The medians of the bounds are shown in the Over and Under columns in Table 2. For all experiments, the baseline failed to generate regions with $> 1000$ examples on most images and, thus, we omit its computed size to save space. On the few images where it does not, the size generated by PARADE is always more.

We report the number of concrete examples, the number of splits, the average runtime on attackable regions, and the number of successfully verified regions for both the baseline and PARADE in the #Splits, Time, and #VerReg columns. Table 2 demonstrates that in a similar or smaller amount of time, PARADE is capable of generating robust adversarial examples for most regions $\mathcal{O}$, while the baseline fails for most. Further, our regions are non-trivial as they contain more than $10^{64}$ concrete adversarial attacks. We note that the runtimes for generating robust examples in this setting is higher than in Table 1. This is due to both the inefficiency of the attack and the time required for the robustness certification of a single region. Due to the flexibility and the generality of our method, we believe it can be easily adapted to more efficient attack and certification methods in the future.

## 5.3 ROBUSTNESS OF ADVERSARIAL EXAMPLES TO RANDOMIZED SMOOTHING

Next, we show that our adversarial examples robust to intensity changes are significantly more effective against state-of-the-art defenses based on randomized smoothing compared to the concrete attacks by $\mathbb{A}$ generated for our examples, as well as the baseline from Table 1. We note that our setting is different than Salman et al. (2019) which directly attacks the smoothed classifier. Instead, we create examples with PARADE on the original network $f$ with $\mathbb{A}$, based on the $L_2$ PGD attack and evaluate them on $g$. For all methods, we calculate the quantity $R_{\text{adv}}$ for $g$, introduced in Definition 2, that measures the adversarial attack strength on $g$. For PARADE and the baseline we select $\tilde{x}$ in Definition 2 as the middle point of the underapproximation box projected back to the $L_2$ ball, with center $x$ and radius the adversarial distance $R'$. For the concrete attacks, $\tilde{x}$ is simply the attack itself. We exclude images whose adversarial distance $R'$ is more than $33\%$ bigger than the certified radius $R$ of $x$, as we consider them robust to attacks. We experiment with networks $f$ with the same architectures as in Table 1. We supply additional details on the experimental setup in Appendix C.4.

In Table 3, we show the ratio between the adversarial strengths $R_{\text{adv}}$ of all other methods and PARADE averaged across the first 100 test images. The last two rows in Table 3 depict the mean and the $95\%$ percentile of the strength ratio across the attacks generated by $\mathbb{A}$. We note the smaller the ratio in Table 3 is, the weaker the attacker is with respect to PARADE. We observe that our examples

produce attacks on the smoothed classifier $g$ with average strength ranging from $2\times$ to $6\times$ larger compared to the alternatives. We conjecture, this is caused by our examples being chosen such that the regions around them have comparatively higher concrete example density. We note the uniform baseline also largely performs better than the mean individual attack, further supporting our claim.

### 5.4 COMPARISON WITH EMPIRICALLY ROBUST EXAMPLES

We compare our provably robust adversarial examples to empirically robust adversarial examples (Athalye et al., 2018) – large input regions, defined by a set of perturbations of an image, that are empirically verified to be adversarial. The empirical verification is based on calculating the expectation over transformations (EoT) of the obtained region defined in Athalye et al. (2018). We consider a region to be an empirically robust adversarial example if it has an EoT $\geq 90\%$. We demonstrate that EoT is a limited measure, as it is not uniform. That is, empirically robust adversarial examples can exhibit high EoT - e.g $95\%$, while specially chosen subregions incur very low EoT scores. In Appendix C.5, we discuss our extension to Athalye et al. (2018) used in this experiment.

We execute the experiment on the first MNIST `ConvBig` transformation in Table 2. We note that the algorithm from Athalye et al. (2018) relies on sampling and does not scale to high dimensional $L_\infty$ perturbations. The algorithm produces 24 different regions, significantly less than our 44 provable regions. The 24 empirically robust adversarial examples produced achieve average EoT of $95.5\%$. To demonstrate the non-uniformness of the empirical robustness, for each region, we considered a subregion of the example, with dimensions of size $10\%$ of the original example, placed at the corner of the example closest to the original unperturbed image. We calculated the EoT of these subregions and found that $14/24$ regions had an EoT of $< 50\%$, with 9 having an EoT of $0\%$.

This demonstrates that the fundamental assumption that the EoT of a region is roughly uniform throughout does not hold and instead it is possible and even common in practice that a significant part of an empirically robust example is much less robust than the remaining region. Further, this often happens close to the original input image, thus providing the misleading impression that an empirically robust adversarial example is a much stronger adversarial attack than it actually is. This misleading behaviour suggests that provably robust adversarial examples, which cannot exhibit such behaviour by construction, provide a more reliable method of constructing adversarial examples invariant to intensity changes and geometric transformations. We note that a limiting factor to the practical adoption of our method remains scalability to real world networks and datasets such as ImageNet, due to the reliance on certification techniques that do not scale there yet. We note that our method is general and efficient enough to directly benefit from any future advancement in verification.

### 5.5 COMPARISON WITH LIU ET AL. (2019)

In this section, we extend the method proposed in Liu et al. (2019) to generate provably robust hyperbox regions against pixel intensity changes and compare the resulting algorithm to PARADE. As the code provided in Liu et al. (2019) only applies to fully connected networks, we applied both their approach and PARADE on a new 8x200 MNIST fully connected network with the same architecture as the one used in Table 1. To facilitate simpler comparison, the network was trained using the code provided in Liu et al. (2019). We initialized both methods with the same overapproximation hyperbox $\mathcal{O}$ and compare the shrinking phase. With similar runtime ($\approx 200$s), out of 84 possible regions, PARADE and Liu et al. (2019) created 84 and 76 regions with median underapproximation volume of $10^{184}$ and $10^{46}$, respectively. We point out that our method is capable of creating polyhedral regions and handle the geometric perturbations, while Liu et al. (2019) cannot.

## 6 CONCLUSION

We introduced the concept of provably robust adversarial examples – large input regions guaranteed to be adversarial to a set of perturbations. We presented a scalable method called PARADE for synthesizing such examples, generic enough to handle multiple perturbations types including pixel intensity and geometric transformations. We demonstrated the effectiveness of PARADE by showing it creates more and larger regions than baselines, in similar or less time, including regions containing $\approx 10^{573}$ adversarial examples for pixel intensity and $\approx 10^{599}$ for geometric transformations.

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

Table 4: Comparison between regions created with different values for the parameter $c$ introduced in Algorithm 1 on the MNIST `ConvBig` Sc(20) T(0,1,0,1) experiment. Columns Under and Over show the median bounds on the number of concrete adversarial examples contained within the regions. The columns Time and #It depict the average time and number of iterations taken by Algorithm 1.

| C | #VERREG | UNDER | OVER | TIME | #IT |
|---|---|---|---|---|---|
| 0.45 | 25 | $> 10^{59}$ | $< 10^{167}$ | 264 s | 1.6 |
| 0.55 | 36 | $> 10^{52}$ | $< 10^{162}$ | 333 s | 2.9 |
| 0.65 | 42 | $> 10^{64}$ | $< 10^{174}$ | 441 s | 4.5 |
| 0.75 | 44 | $> 10^{67}$ | $< 10^{181}$ | 585 s | 6.7 |
| 0.85 | 44 | $> 10^{113}$ | $< 10^{199}$ | 871 s | 10.6 |

## A  ADDITIONAL EXPERIMENTS

### A.1  DEPENDENCE OF PARADE ON THE VALUE OF $c$

In this section, we examine the effect of the parameter $c$, introduced in Algorithm 1, on the size of our provably robust adversarial examples. We show how the number of concrete adversarial examples contained in our regions robust to geometric transformations vary, as a function of $c$. Table 4 depicts the results on the MNIST `ConvBig` scaling and translation experiment, originally depicted in Table 2. The columns Time and #It depict the average time and number of iterations taken by Algorithm 1 in order to converge. Similarly to Table 2, the #VerReg column depicts the number of verified regions. The results correspond to our intuition for $c$: the higher the value of $c$ is, the greater the volume produced by our method and vice versa. Further, we observe that for higher values of $c$, our algorithm succeeds to produce significantly more verified regions. We chose the value $c = 0.65$ for the experiments in Table 2 in the main body of the paper, as it represents a good trade-off between the speed and the number of regions recovered, as shown in Table 4.

## B  FURTHER DETAILS ON OUR ALGORITHM DESIGN

### B.1  ADVERSARIAL EXAMPLE DEFINITIONS

In this section, we provide precise definitions of our adversarial examples robust to pixel intensity changes and geometric changes, used in Table 1 and Table 2.

#### B.1.1  ADVERSARIAL EXAMPLES ROBUST TO $L_\infty$ PIXEL INTENSITY CHANGES

**Definition 4.** *For an image $x \in \mathbb{R}^{n_0}$, a neural network $f$, adversarial budget $\epsilon \in \mathbb{R}$, and adversarial class $y_t \neq f(x)$, a convex region $\mathcal{I} \subseteq \mathbb{R}^{n_0}$ is called an adversarial example around $x$ robust to pixel intensity changes, iff for all points $\bar{x} \in \mathcal{I}$ it is satisfied that $\|\bar{x} - x\|_\infty \leq \epsilon$ and $y_t = f(\bar{x})$.*

#### B.1.2  ADVERSARIAL EXAMPLES ROBUST TO GEOMETRIC CHANGES

**Definition 5.** *Let $\mathcal{T} : \mathbb{R}^{n_0} \times \mathbb{R}^t \to \mathbb{R}^{n_0}$ be a geometric perturbation that takes an image $x \in \mathbb{R}^{n_0}$ and a vector $p \in \mathbb{R}^t$ of geometric parameters, bounded within the hyperbox $[p^l, p^u]$, given by the lower and upper bound vectors $p^l, p^u \in \mathbb{R}^t$, and produces geometrically transformed version of the image, denoted $\bar{x}$, that should be classified the same as $x$. For an image $x \in \mathbb{R}^{n_0}$, a neural network $f$, and adversarial class $y_t \neq f(x)$, a convex region $\mathcal{I} \subseteq \mathbb{R}^t$ is called an adversarial example around $x$ robust to the geometric perturbation $\mathcal{T}$, iff for all points $\bar{p} \in \mathcal{I}$ it is satisfied that $\bar{p} \in [p^l, p^u]$ and $y_t = f(\mathcal{T}(x, \bar{p}))$.*

We note that this definition allows for $\mathcal{T}$ to be a complex composition of simple geometric transformations such as rotations and translations, as it is the case in Table 2. In this case, $p$ is a vector of the parameters of all transformations. We further note that while Definition 5 defines $\mathcal{I}$ in terms of the geometric parameters $p$, one can also look at $\mathcal{I}$, as the region in image space obtained by propagating all possible $\bar{p}$ through $\mathcal{T}$. DeepG Balunović et al. (2019) overapproximates the later region in image space with a polyhedra to produce the polyhedral region $\bar{\mathcal{I}}$ in image space. In Table 2, we report the underapproximation and overapproximation sizes of $\bar{\mathcal{I}}$ in terms of number of concrete images it contains.

---

**Algorithm 2** PGD_PROJECT

---

1: **func** PGD_PROJECT( $l^i$, $u^i$, $l^{i-1}$, $u^{i-1}$, hp, $a$, $b$, $p_{i-1}$ )
2:    $l^i$ = Clip( $l^i$, $l^{i-1}$, $u^{i-1}$ )
3:    $u^i$ = Clip( $u^i$, $l^i$, $u^{i-1}$ )
4:    val = hp( $l^i$, $u^i$ ) $- b$
5:    **if** val $\geq -p_{i-1}$ **then**
6:       **return** $l^i$, $u^i$
7:    **end if**
8:    $[l^i]_{a>0}$ = $-\frac{[l^i]_{a>0} \cdot (p_{i-1} - b)}{\text{val}}$
9:    $[u^i]_{a<0}$ = $-\frac{[u^i]_{a<0} \cdot (p_{i-1} - b)}{\text{val}}$
10:    $l^i$ = Clip( $l^i$, $l^{i-1}$, $u^{i-1}$ )
11:    $u^i$ = Clip( $u^i$, $l^i$, $u^{i-1}$ ) RETURN $l^i$, $u^i$
12: **end func**

---

### B.2 DETAILED DESCRIPTION OF SHRINK_LP

In this section we describe in detail the function SHRINK_LP, used to shrink $\mathcal{U}_{i-1}$ to $\mathcal{U}_i$ when generating examples robust to pixel intensity changes, based on the parameters $L^{i-1}(x)$ and $p_{i-1}$. SHRINK_LP defines two sets of LP variables for the lower bounds $l^i \in \mathbb{R}^{n_0}$ and the upper bounds $u^i \in \mathbb{R}^{n_0}$ of $\mathcal{U}_i$. As described in Section 3.2, the LP optimizes the sum of widths of the dimensions of $\mathcal{U}_i$, i.e., $\sum_{j=1}^{n_0}([u^i]_j - [l^i]_j)$.

Recall that $L^{i-1}(x)$ is a linear function of $x$ and, thus, can be represented as $L^{i-1}(x) = a \cdot x + b$. As described in Section 3.2, the LP we define needs to enforce $\min_{x \in \mathcal{U}_i} L^{i-1}(x) \geq -p_{i-1}$. We rewrite $\min_{x \in \mathcal{U}_i} L^{i-1}(x)$ as:

$$\text{hp}( l^i, u^i ) = \max(0, a) \cdot l^i + \min(0, a) \cdot u^i + b.$$

Since $\text{hp}(l^i, u^i)$ is a linear function of $l^i$ and $u^i$, we add the constraint $\text{hp}(l^i, u^i) \geq -p_{i-1}$ to the LP. This results in the LP problem:

$$\begin{aligned}
\bar{l}^i, \overline{u}^i = \quad & \text{argmax} \sum_{j=1}^{n_0}([u^i]_j - [l^i]_j) \\
& \text{s.t. } \text{hp}(l^i, u^i) \geq -p_{i-1} \\
& \quad [l^{i-1}]_j \leq [l^i]_j \leq [u^i]_j \leq [u^{i-1}]_j \text{ for all } j.
\end{aligned}$$

where $l^{i-1}, u^{i-1} \in \mathbb{R}^{n_0}$ represent the lower and upper bounds of $\mathcal{U}_{i-1}$, the second constraint enforces $\mathcal{U}_i \subseteq \mathcal{U}_{i-1}$ and $\bar{l}^i, \overline{u}^i \in \mathbb{R}^{n_0}$ denotes the LP solution. Finally, SHRINK_LP outputs the hyperbox $\mathcal{U}_i = [\bar{l}^i, \overline{u}^i]$.

### B.3 DETAILED DESCRIPTION OF SHRINK_PGD

In this section we describe in detail the function SHRINK_PGD, used to shrink $\mathcal{U}_{i-1}$ to $\mathcal{U}_i$ for generating regions robust to geometric transformations, based on the parameters $L^{i-1}(x)$ and $p_{i-1}$.

SHRINK_PGD relies on PGD to optimize the log volume of $\mathcal{U}_i$ with respect to its lower and upper bounds, as described in Section 3.2. The optimization problem can be written as:

$$\begin{aligned}
\bar{l}^i, \overline{u}^i = \quad & \text{argmax} \sum_{j=1}^{n_0} \log([u^i]_j - [l^i]_j) \\
& \text{s.t. } \text{hp}(l^i, u^i) \geq -p_{i-1} \\
& \quad [l^{i-1}]_j \leq [l^i]_j \leq [u^i]_j \leq [u^{i-1}]_j \text{ for all } j,
\end{aligned} \tag{1}$$

where we use the same notations as in the previous section. Since we rely on PGD, we need to define how to project values of $l^i$ and $u^i$ that violate the constraints in Equation 1.

Algorithm 2 demonstrates the heuristic we use for the projection. Algorithm 2 slightly abuses the notations above, and treats $l^i$ and $u^i$, as the concrete values of $l^i$ and $u^i$ to be projected, instead of treating them as variables.

It first clips $l^i$ and $u^i$ to enforce $\mathcal{U}_i \subseteq \mathcal{U}_{i-1}$ (Line 2—3). It then checks whether $\text{hp}(l^i, u^i) \geq -p_{i-1}$ is satisfied and if so returns (Line 4—7). Otherwise, all elements of $l^i$ corresponding to positive values

Table 5: Neural networks used in our experiments. For each, we list the number of layers and neurons, as well as, their type (fully connected (FFN) or convolutional (Conv)).

| DATASET | MODEL | TYPE | NEURONS | LAYERS |
|---|---|---|---|---|
| | $8 \times 200$ | FFN | 1,610 | 9 |
| MNIST | CONVSMALL | CONV | 3,604 | 3 |
| | CONVBIG | CONV | 48,064 | 6 |
| CIFAR10 | CONVSMALL | CONV | 4,852 | 3 |
| | CONVBIG | CONV | 62,464 | 6 |

of $a$, $[l^i]_{a>0}$, and all elements of $u^i$ corresponding to negative values of $a$, $[u^i]_{a<0}$, are adjusted by a factor of $-\frac{p_{i-1}-b}{\text{val}}$, where $\text{val} = \text{hp}(l^i, u^i) - b$. This enforces $\text{hp}(l^i, u^i) \geq -p_{i-1}$ (Line 8—9). In rare cases the second projection can result in values of $l^i$ and $u^i$ violating $\mathcal{U}_i \subseteq \mathcal{U}_{i-1}$. Our solution is to clip again (Line 10—11).

After the last clipping operation, in the rare case mentioned above the returned values for $l^i$ and $u^i$ violate $\text{hp}(l^i, u^i) \geq -p_{i-1}$. However, we did not find this to be a problem in practice, as it occurs rarely. Further, one can view such violations, as choosing a different value for $c$ in Algorithm 1, resulting in different value for $-p_{i-1}$.

## C    FURTHER DETAILS ON EXPERIMENTAL SETUP

### C.1    NETWORKS

Table 5 shows the architectures of the networks on which we evaluate.

### C.2    FURTHER DETAILS ON THE EXPERIMENTAL SETUP IN SECTION 5.1

In this section, we provide further details on the experimental setup in Section 5.1. In this experiment, for creating $\mathcal{O}$ we use 2500 adversarial examples each from the Frank-Wolfe optimization algorithm (Frank & Wolfe, 1956) and PGD with step size $0.1\epsilon$ and $0.01\epsilon$, respectively. For the PGD examples, we use output diversification (Tashiro et al., 2020) with 5 iterations. Throughout the experiment, we set $c = 0.99$ and multiply it by a factor of $0.99$ at each iteration of Algorithm 1.

### C.3    FURTHER DETAILS ON THE EXPERIMENTAL SETUP IN SECTION 5.2

In this section, we provide further details on the experimental setup in Section 5.2. For all experiments in Table 2, we compute $\mathcal{O}$ using random sampling attacks which are the state-of-the-art for geometric transformations (Engstrom et al., 2019). For the 3 dimensional experiments, we rely on 15000 random samples, while for the 4D configurations we use 50000. We use $c = 0.65$ for all experiments, except for the 4 dimensional CIFAR10 experiment, where we use $c = 0.75$ to increase the precision of PARADE. For the MNIST experiments in Table 2, we execute SHRINK_PGD with 50 initialization and 200 PGD steps of size $5 \times 10^{-5}$. For CIFAR10 experiments, we use 500 initialization and 50 PGD steps instead.

### C.4    DETAILED DESCRIPTION OF EXPERIMENTAL SETUP IN SECTION 5.3

***Neural networks***    In Section 5.3, we experiment with networks $f$ with the same architectures as in Table 5. Following the standard practices for training networks used alongside smoothing, outlined in Cohen et al. (2019), we trained the networks on images with added Gaussian noise with a standard deviation of $0.15$.

***Selecting $\alpha$ and $\sigma$***    In the experiments in Section 5.3, we need to select $\alpha$ and $\sigma$ to compute $R_{\text{adv}}$ using Definition 2. For all methods we use $\alpha = 0.005$, as suggested by Cohen et al. (2019). We select the $\sigma$ value that produces the biggest adversarial strength $R_{\text{adv}}$ for our method and use it for computing $R_{\text{adv}}$ for all methods. We note that tuning $\sigma$ for the individual attacks generated by $\mathbb{A}$ is very computationally intensive and therefore we avoid it.

**Selecting** $R'$    In this paragraph, we detail how the adversarial distance $R'$ is chosen. We choose it heuristically, for each individual $x$. For each $x$, we select $R'$ by searching for the smallest adversarial distance on $f$, where at least 10% of 500 attacks with $\mathbb{A}$ on $f$ succeed but exclude images $x$, whose $R'$ is more than 33% bigger than the certified radius $R$ of $g$. This procedure allows us to select the adversarial distance $R'$, that is neither too small, nor too big, which is important to avoid making the problem trivial, as outlined in Section 2.3. In particular, for this choice of $R'$ the smoothed classifier $g$ is likely to be attackable, since $f$ contains enough attacks for a high density input adversarial region to exist. On top of that, the exclusion of images $x$ with too big adversarial distance $R'$ experimentally allowed us to exclude trivial attacks, for which $x$ is attackable for most classes on $g$.

## C.5   DETAILED DESCRIPTION OF ADAPTED EoT USED IN EXPERIMENTS IN SECTION 5.4

In this section, we discuss the extension of Athalye et al. (2018) we use to produce the empirically robust adversarial examples for the experiment in Section 5.4. We define an optimization procedure, where we seek a hyperbox geometric region $\mathcal{U}_{\text{emp}} \subseteq \mathbb{R}^{n_0}$, defined in terms of the lower bound $l_{\text{emp}} \in \mathbb{R}^{n_0}$ and the upper bound $u_{\text{emp}} \in \mathbb{R}^{n_0}$ vectors. We seek to find $\mathcal{U}_{\text{emp}}$ within the set of allowed geometric transformations $\mathcal{I} \subseteq \mathbb{R}^{n_0}$, such that $\mathcal{U}_{\text{emp}}$ is an empirically robust adversarial example with maximal log volume, and high EoT. Similarly to Athalye et al. (2018) EoT is calculated on set of samples from $\mathcal{U}_{\text{emp}}$. We rely on differentiable bilinear interpolation (Jaderberg et al., 2015) to allow gradients to flow through the geometric transformations we consider and optimize $l_{\text{emp}}$ and $u_{\text{emp}}$ using PGD.

## D   ANALYSIS OF ALGORITHM 1

In this section, we analyze the properties of PARADE.

### D.1   SOUNDNESS OF ALGORITHM 1

**Theorem 1.** *If Algorithm 1 converges, it returns a provably robust region $\mathcal{U}$.*

*Proof.* Algorithm 1 can only complete its execution using Line 6 or Line 18. Since we assume the algorithm successfully converged, returning at Line 18 is impossible. Therefore, the algorithm must have returned the region $\mathcal{U}_{i-1}$ at Line 6 and, therefore, $e^{i-1}$ must be non-negative (otherwise we would not have taken the right branch at Line 5). However, $e^{i-1}$ is obtained by executing the verifier $\mathbb{V}$ on $\mathcal{U}_{i-1}$ at Line 4. By the definition of our verifier $\mathbb{V}$, it returns non-negative $e^{i-1}$ only if $\mathcal{U}_{i-1}$ is provably robust. This finishes our proof. □

### D.2   CONVERGENCE OF ALGORITHM 1

In this subsection, we demonstrate that Algorithm 1 converges exponentially fast under a suitable assumption for the monotonicity of the neural network certification method $\mathbb{V}$. To this end, we introduce the following definition of monotonicity of $\mathbb{V}$, that we will leverage for the rest of the subsection.

**Definition 6.** *Let* $\mathbb{V}(f, \mathcal{I}, y_t)$ *be a neural network certification method that returns the certification objective* $L^{\mathcal{I}}(x)$ *for an input region* $\mathcal{I}$ *and target* $y_t$ *on the neural network* $f$. *We call* $\mathbb{V}$ *objective-monotonic if for all* $f$ *and input regions* $\mathcal{X}$ *and* $\mathcal{Y}$ *with* $\mathcal{X} \subseteq \mathcal{Y}$ *the property:*

$$\min_{x \in \mathcal{X}} L^{\mathcal{X}}(x) \geq \min_{x \in \mathcal{X}} L^{\mathcal{Y}}(x)$$

*is satisfied.*

Intuitively, Definition 6 states that a verifier is objective-monotonic if the certification objective produced for a smaller region $\mathcal{X}$ is always higher (closer to verification) than the certification objective produced for a bigger region $\mathcal{Y}$, when both of them are optimized over the smaller region $\mathcal{X}$.

We note that most practical neural network certification methods, including DeepPoly, are not objective-monotonic — that is one can find triples $f$, $\mathcal{X}$ and $\mathcal{Y}$ violating the definitions above.

However, in practice we find that for all practical neural network certification methods the definitions hold for most triples. This is expected, as for violating triples we know that the overapproximation generated for $\mathcal{X}$ needs to be larger than the one generated for $\mathcal{Y}$. Intuitively, if that was the case for a large portion of the triples, the certification method will not be tight.

Next, we leverage this defition to demonstrate that our algorithm converges exponentially fast under the idealized condition that the certification method used is objective-monotonic.

**Theorem 2.** *For objective-monotonic certification methods $\mathbb{V}$ in Algorithm 1, the certification error at the $i$-th iteration $e^i$ converges exponentially fast towards $0$ (towards certification).*

*Proof.* To prove Theorem 2, we will leverage the definition of objective-monotonic certification method above to bind the certification error between consequitive iterations of Algorithm 1. We will then use the bound to derive a bound on the certification error $e^i$ in terms of the intial certification error $e^0$. Finally, we will use the bound to show the exponential convergence.

Since $\mathcal{U}_i \subseteq \mathcal{U}_{i-1}$, we can apply Definition 6 to get the inequality:

$$\min_{x \in \mathcal{U}_i} L^i(x) \geq \min_{x \in \mathcal{U}_i} L^{i-1}(x).$$

By construction of $\mathcal{U}_i$, we further know:

$$\min_{x \in \mathcal{U}_i} L^{i-1}(x) \geq -p_{i-1} = e^{i-1} \cdot c.$$

Combining both inequalities and noting that $e^i = \min_{x \in \mathcal{U}_i} L^i(x)$, we get:

$$e^i \geq e^{i-1} \cdot c. \tag{2}$$

Since Equation 2 holds for all $i$, we can apply it recursively on its right-hand side to get:

$$e^i \geq e^0 \cdot c^i. \tag{3}$$

We recall that $e^i$ is negative unless $\mathcal{U}_i$ verifies. Therefore by Equation 3, $e^i$ converges exponentially fast towards $0$, regardless of $e^0$. $\qquad\square$

## E  PROVABLY ROBUST POLYHEDRA ADVERSARIAL EXAMPLES

In this section, we show an extension to PARADE that allows us to generate provably robust adversarial examples represented as a general polyhedron shapes $\mathcal{P}$ in the neural network input space. The approach relies on the underapproximation and overapproximation boxes $\mathcal{U}$ and $\mathcal{O}$, presented in the main body, to guide the search of $\mathcal{P}$ and is demonstrated in Figure 2. The results of the method presented in this section were shown in Table 1 in Section 5.1.

### E.1  BACKGROUND

Our method for generating polyhedral robust adversarial examples requires more restricted form for $\mathbb{V}$, compared to the algorithms presented in the main body. In this subsection, we discuss the relevant background and the additional requirements we pose on $\mathbb{V}$ and introduce the relevant notation for the rest of this section.

For this section, we will focus on ReLU-based neural networks. We note that our concepts can easily be extended to networks with other activation functions such as sigmoid or tanh. We consider neural network with $l$ layers $f_1, \ldots, f_l$. The first $l-1$ layers are of the form $f_i(\mathbf{x}) = \max(0, A_i \cdot \mathbf{x} + \mathbf{b}_i)$ for $i \in \{1, \ldots, l-1\}$ and the final layer is of the form $f_l(\mathbf{x}) = A_l \cdot \mathbf{x} + \mathbf{b}_l$. The neural network $f$ is the composition of all layers: $f = f_l \circ \cdots \circ f_1$. Let $n_i$ be the number of neurons in layer $i$. If $z^a_{i,j}$ is the input to the $j$-th ReLU activation in layer $i$ and $z^r_{i,j}$ is the corresponding ReLU output, we can represent the neural network as the following system of constraints: For $i \in \{1, \ldots, l\}$ and $j \in \{1, \ldots, n_i\}$,

$$z^a_{i,j} = [b_i]_j + \sum_{k=1}^{n_{i-1}} [A_i]_{j,k} \cdot z^r_{i-1,k},$$

$$z^r_{i,j} = \max(0, z^a_{i,j}).$$

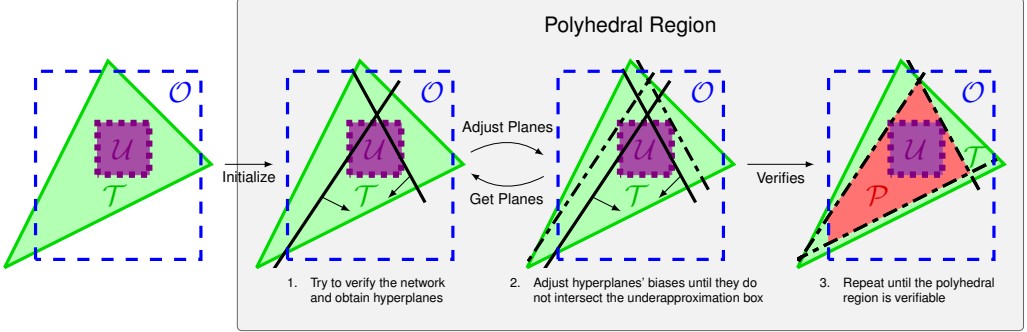

Figure 2: Overview of our method for generating provably robust polyhedral adversarial examples. The green triangle denoted $\mathcal{T}$ represents the ground truth adversarial region. The dashed blue rectangle denoted $\mathcal{O}$ and the violet dotted rectangle denoted $\mathcal{U}$ represent the fitted overapproximation and underapproximation boxes, respectively. The solid black lines represent hyperplanes generated by the certification procedure $\mathbb{V}$, while their dash-dotted counterparts represent the hyperplanes after bias-adjustment. The small arrows going out of the hyperplanes in Step 1 and Step 2 represent the direction which is retained by the hyperplanes' corresponding half-spaces. The output polyhedral region denoted $\mathcal{P}$ is shown in red in Step 3.

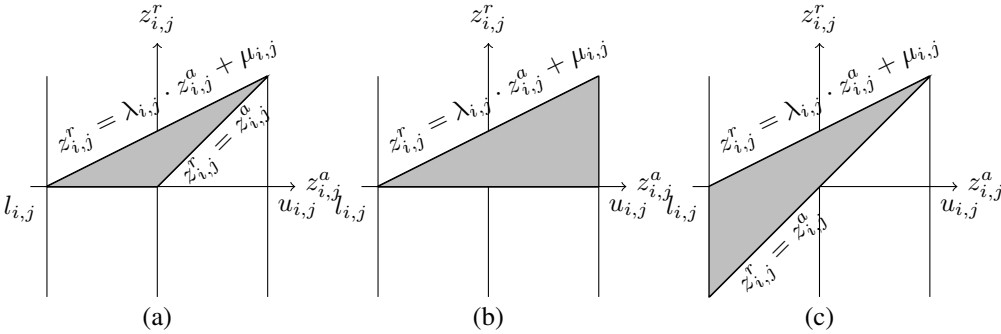

Figure 3: Convex approximations for the ReLU function: (a) shows the triangle approximation Ehlers (2017) with the minimum area in the input-output plane, (b) and (c) show the two convex approximations used in DeepPoly Singh et al. (2019). In the figure, $\lambda_{i,j} = u_{i,j}/(u_{i,j} - l_{i,j})$ and $\mu_{i,j} = -l_{i,j} \cdot u_{i,j}/(u_{i,j} - l_{i,j})$. The figure is taken from Singh et al. (2019).

Here, $z_{0,j}^r$ describes the $j$-th of $n_0$ input activations. We call the two types of constraints affine constraints and ReLU constraints, respectively. We additionally constrain our inputs $z_0^r$ to be points inside of a polyhedron $\mathcal{P} \subseteq \mathbb{R}^{n_0}$. An adversarial class $y_t$ has the highest score on all inputs in $\mathcal{P}$ if $\min(z_{l,y_t}^a - z_{l,y}^a) > 0$ with respect to all constraints for all $y \in \{1, \ldots, n_l\} \setminus \{y_t\}$. The goal in section will be to find a large polyhedron $\mathcal{P}$ for which we can verify that this is true.

***Linear approximations of ReLU*** As reasoning about neural network constraints directly is often intractable, given bounds $l_{i,j} \leq z_{i,j}^a \leq u_{i,j}$, the *triangle approximation* (Ehlers, 2017) relaxes the ReLU constraints such that all involved inequalities are linear:

$$z_{i,j}^r \geq z_{i,j}^a, \quad z_{i,j}^r \geq 0,$$
$$z_{i,j}^r \leq \lambda_{i,j} \cdot z_{i,j}^a + \mu_{i,j}.$$

Figure 3 (a) visualizes the triangle approximation. Here, $\lambda_{i,j} = u_{i,j}/(u_{i,j} - l_{i,j})$ and $\mu_{i,j} = -l_{i,j} \cdot u_{i,j}/(u_{i,j} - l_{i,j})$ are selected such that this set of constraints describes the convex hull of the ReLU constraints in the $(z_{i,j}^a, z_{i,j}^r)$-plane and the given bounds.

The DeepPoly approximation (Singh et al., 2019) further relaxes the triangle approximation by keeping only one of the lower bounds on each variable $z_{i,j}^r$. It picks either $z_{i,j}^r \geq z_{i,j}^a$ or $z_{i,j}^r \geq 0$, whichever minimizes the area of the resulting triangle in the $(z_{i,j}^a, z_{i,j}^r)$-plane. Figure 3 (b) and (c) show the two convex relaxations used in DeepPoly.

We can obtain $l_{i,j}$ and $u_{i,j}$ by optimizing each component of $z_{i,j}^a$ according to the previously determined constraints for layers $0, \ldots, i$. We note that in the special case where $u_{i,j} \leq 0$ or $l_{i,j} \geq 0$, no approximation is needed and $z_{i,j}^r$ is directly assigned $0$ or $z_{i,j}^a$, respectively. Neurons for which the approximation is needed are called *undecided*.

***Computing neuron bounds*** Most verification algorithms based on convex relaxation start with a chosen input region for which they calculate the bounds $l_{i,j}$ and $u_{i,j}$ of $z_{i,j}^a$ in layer-by-layer fashion, where the constraints for previous layers are used to derive the bounds for layer $i$. In addition to DeepPoly and the triangle approximation, this is also true for other commonly used convex relaxation algorithms such as CROWN (Zhang et al., 2018) and DeepZ (Singh et al., 2018a).

Next, we briefly describe how the bounds are computed for a particular layer in DeepPoly. We focus on DeepPoly since it is the particular convex relaxation algorithm we chose to employ in this section. For all neurons, DeepPoly stores a single linear lower and upper bound inequality. These inequalities bound the neuron's value in terms of the values of neurons from previous layers in the network. For ReLU neurons, these are the inequalities described above. For affine neurons, these are simply the affine transformations, where the affine equality is converted to a pair of inequalities.

To arrive at the bounds $l_{i,j}$ and $u_{i,j}$ of $z_{i,j}^a$, DeepPoly repeatedly backsubstitutes variables in the linear inequalities of $z_{i,j}^a$ with their corresponding linear inequalities from previous layers. The process continues until we arrive at linear inequalities $L_{i,j}(z_0^r) \leq z_{i,j}^a \leq U_{i,j}(z_0^r)$, with $L_{i,j}(x) = \underline{a}_{i,j} \cdot x + \underline{b}_{i,j}$ and $U_{i,j}(x) = \overline{a}_{i,j} \cdot x + \overline{b}_{i,j}$ denoting linear functions in terms of input variables $x \in \mathbb{R}^{n_0}$ with coefficients $\underline{a}_{i,j}, \overline{a}_{i,j} \in \mathbb{R}^{n_0}$ and $\underline{b}_{i,j}, \overline{b}_{i,j} \in \mathbb{R}$. The final bounds $l_{i,j}$ and $u_{i,j}$ are obtained by optimizing the respective functions $L_{i,j}(x)$ and $U_{i,j}(x)$ over the input region.

We note that our algorithm does not depend on details of DeepPoly's backsubstitution algorithm: It only requires the existence of linear functions $L_{i,j}(x)$ and $U_{i,j}(x)$ bounding neuron values $z_{i,j}^a$ in terms of input activations. This makes it compatible with other popular verification algorithms based on convex relaxation, such as CROWN (Zhang et al., 2018) and DeepZ (Singh et al., 2018a).

***Certification Objective*** To certify the robustness of an input region, $\mathbb{V}$ needs to certify that $\min(z_{l,y_t}^a - z_{l,y}^a) > 0$ for all $y \in \{1, \ldots, n_l\} \setminus \{y_t\}$. To calculate lower bounds on those minima, a common trick is to introduce an affine layer that computes the objectives $z_{l,y_t}^a - z_{l,y}^a$ for each $y$. We will refer to this layer as objective layer. To obtain lower bounds on our objectives, we compute lower bounds of the corresponding neuron activations in the objective layer with the usual backsubstitution procedure.

For the rest of the section we will denote the backsubstituted linear lower bounds on our objectives in terms of input activations for a given target $y_t$ with $L_{l+1,y}(x) = \underline{a}_{l+1,y} \cdot x + \underline{b}_{l+1,y}$ for each $y$. Finally, to verify that $\min(z_{l,y_t}^a - z_{l,y}^a) > 0$, it suffices to show that $L_{l+1,y}(x) > 0$ for each point $x$ in the input region and all $y \neq y_t$. We note that the function denoted $L_{l+1,y}(x)$ in this section corresponds to the function denoted $L_y(x)$ in the main body.

### E.2 COMPUTING $\mathcal{P}$

The algorithm for computing $\mathcal{P}$ is presented in Algorithm 3. It takes as input:

- The neural network $f$, as described in Section E.1.
- The underapproximation hyperbox $\mathcal{U}$.
- The overapproximation hyperbox $\mathcal{O}$.
- A neural network certification method $\mathbb{V}(\, f,\, \mathcal{P}_{\text{it}-1},\, i,\, j\,)$ that takes a neural network $f$, a polyhedron $\mathcal{P}_{\text{it}-1}$ at iteration $\text{it} - 1$, the layer number $i$ and the neuron number $j$ and returns the linear functions $L_{i,j}(x)$ and $U_{i,j}(x)$ and the corresponding concrete bounds $l_{i,j}$ and $u_{i,j}$ (See Section E.1).
- The adversarial attack's target class $y_t$.
- The maximum number $p_{\text{it}}$ of iterations used to generate the polyhedron $\mathcal{P}$.

The algorithm computes $\mathcal{P}$, such that $\mathcal{U} \subseteq \mathcal{P} \subseteq \mathcal{O}$ is satisfied. We motivate this choice by noting that we know that $\mathcal{U}$ is certifiably robust and it is therefore a lower bound on $\mathcal{P}$. The algorithm generates

---

**Algorithm 3** GEN_POLY

---

1: **func** GEN_POLY( $f$, $\mathcal{U}$, $\mathcal{O}$, $\mathbb{V}$, $y_t$, $p_{it}$ )
2:      $\mathcal{P}_0 = \mathcal{O}$
3:      **for** it $\in \{1, 2, \ldots, p_{it}\}$ **do**
4:          hs$_o$, Ver = Gen_Obj_Planes( $f$, $\mathcal{U}$, $\mathcal{P}_{it-1}$, $\mathbb{V}$, $y_t$ )
5:          hs$_b$ = Gen_Bound_Planes( $f$, $\mathcal{U}$, $\mathcal{P}_{it-1}$, $\mathbb{V}$ )
6:          $\mathcal{P}_{it}$ = Add_Planes( $\mathcal{P}_{it-1}$, hs$_b$ $\cup$ hs$_o$ )
7:          **if** Ver **then**
8:             **return** $\mathcal{P}_{it-1}$
9:          **end if**
10:      **end for**
11:      **return** FailedToConverge
12: **end func**

---

**Algorithm 4** GEN_OBJ_PLANES

---

1: **func** GEN_OBJ_PLANES( $f$, $\mathcal{U}$, $\mathcal{P}_{it-1}$, $\mathbb{V}$, $y_t$ )
2:      hs$_o$ = []
3:      Ver = True
4:      **for** $y \in \{1, \ldots, n_l\} \setminus \{y_t\}$ **do**
5:          $L_{l+1,y}(x)$, _, $l_y$, _ = $\mathbb{V}$( $f$, $\mathcal{P}_{it-1}$, $l+1$, $y$ )
6:          **if** $l_y \leq 0$ **then**
7:             Ver = False
8:             $\tilde{L}_{l+1,y}(x)$ = Adjust_Bias( $\mathcal{U}$, $L_{l+1,y}(x)$ )
9:             hs$_o$ += [ $\tilde{L}_{l+1,y}(x) \geq 0$ ]
10:          **end if**
11:      **end for**
12:      **return** hs$_o$, Ver
13: **end func**

---

a sequence of polyhedra $\mathcal{P}_0 \supseteq \mathcal{P}_1 \supseteq \ldots \supseteq \mathcal{P}_p$ with $\mathcal{P}_0 = \mathcal{O}$. At each iteration, it computes two sets of half-space constraints – hs$_o$(Line 4) and hs$_b$(Line 5), corresponding to the neurons in the objective and the affine layers of the network. We detail how these constraints are computed in the next two paragraphs. We add these constraints to the polyhedron from the last iteration $\mathcal{P}_{it-1}$ to obtain $\mathcal{P}_{it}$ (Line 6). The algorithm terminates when $\mathcal{P}_{it-1}$ is certified to be robust (Line 7 – 9) and $\mathcal{P}_{it-1}$ is returned. The result of our algorithm is represented by the red region denoted as $\mathcal{P}$ in Step 3 in Figure 2. We note that, since $\mathcal{P}_0 = \mathcal{O}$, $\mathcal{P} \subseteq \mathcal{O}$ is guaranteed by construction, as each subsequent iteration of the algorithm shrinks the polyhedron further. However, $\mathcal{U} \subseteq \mathcal{P}$ depends on the choice of the half-space constraints.

***Generating Certification Objective Constraints***     Algorithm 4 details the procedure we use to collect the half-space constraints corresponding to the objective layer of the network. To obtain constraints from $\mathbb{V}$, we first compute the linear functions $L_{l+1,y}(x)$ and their respective certification errors $l_y$ for all output classes $y \in \{1, \ldots, n_l\} \setminus \{y_t\}$ (Line 5). We know that adding the constraints $L_{l+1,y}(x) \geq 0$ to our polyhedron $\mathcal{P}_{it-1}$ will result in a certifiably robust region $\mathcal{P}_{it}$, however the region will not obey $\mathcal{U} \subseteq \mathcal{P}_{it}$. Therefore, we instead adjust the bias of constraints, so that the resulting hyperplanes $\tilde{L}_{l+1,y}(x) = 0$ do not intersect $\mathcal{U}$ (Line 8). This process is depicted geometrically in Step 2 in Figure 2 and will be described in more detail below. All resulting bias-adjusted hyperplanes are collected in the set hs$_o$ (Line 9) and returned. We note that if $l_y > 0$ for some $y$, the half-spaces $L_{l+1,y}(x) \geq 0$ are trivial. That is $\{L_{l+1,y}(x) \geq 0\} \cap \mathcal{P}_{it-1} = \mathcal{P}_{it-1}$, and thus they are filtered out of hs$_o$ (Line 6). In addition to detecting the trivial constraints, we use the certification errors $l_y$ to check if $\mathcal{P}_{it-1}$ is certifiably robust and return the result alongside hs$_o$ in Line 11.

***Issues Related to Using Only Objective Constraints***     In our experiments, we found that only adding bias-adjusted objective constraints leads to slow convergence of our algorithm. In particular, we found that the first few iterations of the algorithm result in a substantial improvement in the certification objective. However, after this initial fast progress, we observe that the change in the certification objective becomes very small. The reason for this is that the generated constraints from consecutive executions of $\mathbb{V}$ are very similar. We conjecture that the underlying reason is that the constraints no longer change the input shape enough to substantially change the convex approximation of the

---

**Algorithm 5** GEN_BOUND_PLANES

---
1: **func** GEN_BOUND_PLANES( $f$, $\mathcal{U}$, $\mathcal{P}_{\text{it}-1}$, $\mathbb{V}$ )
2:     hs $_b$ = []
3:     **for** $i \in \{1, 2, \ldots, l-1\}$ **do**
4:        **for** $j \in \{1, 2, \ldots, n_i\}$ **do**
5:           $L_{i,j}(x), U_{i,j}(x), l_{i,j}, u_{i,j} = \mathbb{V}(f, \mathcal{P}_{\text{it}-1}, i, j)$
6:           **if** $l_{i,j} \leq 0$ **and** $u_{i,j} \geq 0$ **then**
7:              $\tilde{L}_{i,j}(x) = $ Adjust_Bias$(\mathcal{U}, L_{i,j}(x))$
8:              $\tilde{U}_{i,j}(x) = $ Adjust_Bias$(\mathcal{U}, -U_{i,j}(x))$
9:              hs $_b$ $+=$ $[\,\tilde{L}_{i,j}(x) \geq 0, \tilde{U}_{i,j}(x) \geq 0\,]$
10:           **end if**
11:        **end for**
12:     **end for**
13:     **return** hs $_b$
14: **end func**

---

network. This is in contrast to what we observe when computing the underapproximation hyperbox region $\mathcal{U}$. The difference between the two settings comes from the number of added constraints per iteration. For $\mathcal{U}$, usually many lower and upper bounds are adjusted each iteration, while the objective constraints added are much less, especially for the pixel intensity changes experiment, where the input is very high-dimensional.

***Generating Constraints from Affine Layers***    To facilitate faster convergence, we generate additional constraints for all undecided neurons in the network. We generate two constraints, one with respect to the lower bound of the neuron $L_{i,j}(x) \geq 0$ and one with respect to the upper bound $U_{i,j}(x) \leq 0$. Algorithm 5 describes the process. $L_{i,j}(x)$ and $U_{i,j}(x)$ are computed by $\mathbb{V}$ (Line 5) alongside their concrete bounds $l_{i,j}$ and $u_{i,j}$. The concrete bounds are used to check which neurons are undecided (Line 6).

The generated constraints have the effect of making the neurons decided, while also enforcing substantial change in the convex relaxation of the neural network. Note that the newly obtained constraints are bias-adjusted similar to the objective constraints described above (Line 7–8). We point out that the bias adjustment described below takes care of the problem that $L_{i,j}(x) \geq 0$ and $U_{i,j}(x) \leq 0$ cannot be simultaneously true. Empirically, these hyperplanes serve an important role for improving the convergence rate of our algorithm. Algorithm 5 collects them in the set hs $_b$ (Line 9) and returns them to be added to $\mathcal{P}_{\text{it}-1}$.

***Hyperplane bias adjustment***    So far, we have discussed how constraints are generated but not how we adjust their biases. The bias adjustment algorithm is given in Algorithm 6. The bias adjustment for a lower bound constraint $h(x) \geq 0$ enforces that the intersection of the constraint and the given underapproximation hyperbox $\mathcal{U}$ is $\mathcal{U}$. It does so by computing $h_{\min} = \min_{x \in \mathcal{U}}(h(x))$(Line 2) and changing the given constraint to $h(x) \geq h_{\min}$ (Line 4). The optimization $h_{\min} = \min_{x \in \mathcal{U}}(h(x))$ is analytically solvable, as demonstrated Section B.2. Note that our algorithm adjusts only the constraints with negative $h_{\min}$ (Line 3), since constraints with positive $h_{\min}$ already satisfy the property. The following lemma shows the soundness of our bias-adjustment algorithm:

**Lemma 3.** *The intersection of the bias-adjusted half-space constraint $\tilde{h}(x) \geq 0$ defined in Algorithm 6 with $\mathcal{U}$ is $\mathcal{U}$.*

*Proof.* By the construction of $h_{\min}$, for all points $x \in \mathcal{U}$ we have $h(x) \geq h_{\min}$. We distinguish two cases for the value of $h_{\min}$ — non-negative and negative.

If $h_{\min} \geq 0$, the bias adjustment $b_{\text{adj}}$ is assigned to 0 and $\tilde{h}(x) = h(x)$. Since $h(x) \geq h_{\min}$ for all points $x \in \mathcal{U}$, $\tilde{h}(x) = h(x) \geq h_{\min} \geq 0$ for all points $x \in \mathcal{U}$. Therefore, there does not exist point in $\mathcal{U}$ for which the bias-adjusted half-space constraint $\tilde{h}(x) \geq 0$ is violated. If $h_{\min} < 0$, the bias adjustment $b_{\text{adj}}$ is assigned to $h_{\min}$ and $\tilde{h}(x) = h(x) - h_{\min}$. Therefore, $\tilde{h}(x) = h(x) - h_{\min} \geq h_{\min} - h_{\min} = 0$ for all points $x \in \mathcal{U}$. Analogously to the other case,

---

**Algorithm 6** ADJUST_BIAS

---
1: **func** ADJUST_BIAS($\mathcal{U}$, $h(x)$ )
2:     $h_{\min} = \min\limits_{x \in \mathcal{U}}(h(x))$
3:     $b_{\text{adj}} = \min(0, h_{\min})$
4:     $\tilde{h}(x) = h(x) - b_{\text{adj}}$
5:     **return** $\tilde{h}(x)$
6: **end func**

---

$\tilde{h}(x) \geq 0$ is not violated by the points in $\mathcal{U}$. Since in both cases $\tilde{h}(x) \geq 0$ is not violated by any points in $\mathcal{U}$, the intersection of the constraint and $\mathcal{U}$ is simply $\mathcal{U}$. $\qquad \square$

The bias adjustment for upper bound constraints $h(x) \geq 0$ is achieved by calling Algorithm 6 with $-h(x)$, as in Line 8 in Algorithm 5.

***Adapting DeepPoly to Polyhedral Input Regions***   Out of the box, DeepPoly only handles certification of hyperbox input regions. This is because in order to obtain the concrete bounds $l_{i,j} = \min_{x \in \mathcal{I}} L_{i,j}(x)$ and $u_{i,j} = \max_{x \in \mathcal{I}} U_{i,j}(x)$ for an input region $\mathcal{I} \subseteq \mathbb{R}^{n_0}$ it uses the analytical solution of the optimization problems discussed in Section B.2. In order to use DeepPoly for polyhedral region $\mathcal{I}$, we solve the optimization problem using an Linear Program (LP). This is computationally expensive – it involves solving two LP instances per neuron. For large networks, this can be prohibitive.

To alleviate this issue, in the first iteration of Algorithm 3, we remember which neurons are undecided. The computation of the undecided neurons in the first iteration can be done using regular DeepPoly, as $\mathcal{P}_0 = \mathcal{O}$. In future iterations we solve the LP instances only for the undecided neurons, while using the $\mathcal{P}_0$ bounds for the decided neurons. This optimization is sound, since we have $\mathcal{P}_{\text{it}} \subseteq \mathcal{P}_0$ by construction. This improves the performance substantially – for most certifiably robust input regions in ReLU-based networks we have just a few undecided neurons.

***Proving that $\mathcal{P}$ constraints $\mathcal{U}$ and is contained within $\mathcal{O}$***   In this paragraph, we present a proof for the desired property of Algorithm 3 that $\mathcal{P}$ constraints $\mathcal{U}$ and is contained within $\mathcal{O}$.

**Theorem 4.** *For all polyhedral regions $\mathcal{P}$ generated by Algorithm 3, $\mathcal{U} \subseteq \mathcal{P} \subseteq \mathcal{O}$ holds.*

*Proof.* To prove $\mathcal{P} \subseteq \mathcal{O}$, we leverage the fact that the polyhedra $\mathcal{P}$ is constructed from the overapproximation box $\mathcal{O}$ by intersecting it with the half-space constraints $hb_b^i$ and $hb_o^i$. Since all constraints only reduce the volume of the polyhedra, it follows by induction over the generated half-space constraints that $\mathcal{P}_i \subseteq \mathcal{O}$ for all $i$. Therefore, $\mathcal{P} \subseteq \mathcal{O}$.

To prove $\mathcal{U} \subseteq \mathcal{P}$, we note that the generated half-space constraints $hb_b^i$ and $hb_o^i$ are chosen such that $hb_b^i \cap \mathcal{U} = \mathcal{U}$ and $hb_o^i \cap \mathcal{U} = \mathcal{U}$ for all $i$ (See Lemma 3). By applying this property by induction over the generated half-space constraints, it follows that $\mathcal{U} \subseteq \mathcal{P}_i$ for all $i$. Therefore, $\mathcal{U} \subseteq \mathcal{P}$. $\qquad \square$

## F   VISUALISATION OF ROBUST ADVERSARIAL EXAMPLES

### F.1   DISCUSSION

Figure 4 shows an adversarial region provably robust to intensity changes containing $10^{284}$ adversarial images produced by our method on MNIST `ConvBig`. The original image is of the digit 5, and all images in our region are classified as 3 by the network. The colorbar in Figure 4 quantifies the number of values each pixel can take in our inferred region. The yellow and violet colors represent the two extremes. The intensity of the yellow-colored pixels can vary the most, thus these pixels contribute to more adversarial examples. The intensity of the purple-colored varies the least, thus the adversarial examples in our region are sensitive to the intensity values of these pixels. In our region, the intensity of most background pixels on the edges of the image can vary a lot, as these are green. Violet and green color are more evenly distributed among pixels closer to the foreground (part of the digit "5"). Further, the intensity of several pixels in the foreground can also vary significantly. We

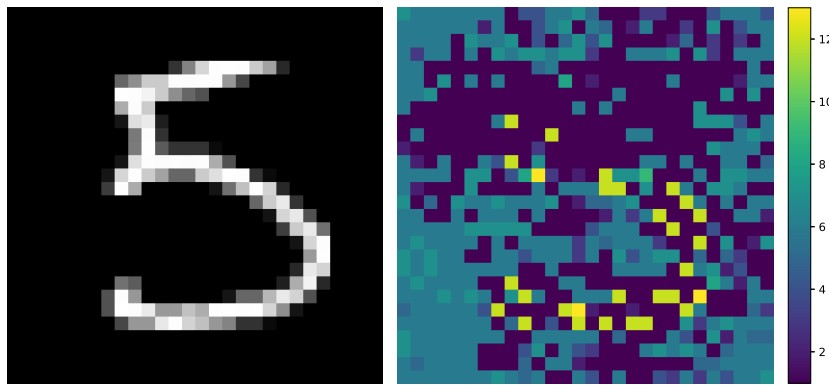

Figure 4: Adversarial region robust to pixel intensity changes.

note that all yellow pixels occur in the foreground. In summary, our region can capture examples that can be generated by significant variations in the intensities of several background and foreground pixels. We supply visualizations for all of the experiments described in Section 5.1 and 5.2 in the next two sections.

## F.2 ADVERSARIAL EXAMPLES ROBUST TO INTENSITY CHANGES

In Figure 5 – 8, we visualise adversarial examples provably robust to pixel intensity changes for the different networks in Table 1. For all figures, the colorbar on the right-hand side quantifies the number of values each pixel can take in our inferred region. The yellow and violet colors represent the two extremes. The intensity of the yellow-colored pixels can vary the most, thus these pixels contribute more to the adversarial examples.

In Figure 5, we show adversarial examples for the MNIST `ConvBig` network. The 3 sub-figures represent the digits 9, 5, and 3, but the images in our regions are classified as 4, 3, and 8, respectively. Our regions are of size $10^{220}$, $10^{284}$, and $10^{262}$, respectively.

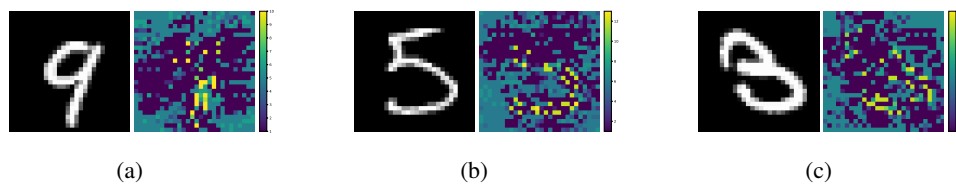

| (a) | (b) | (c) |

Figure 5: Sensitivity of probably robust adversarial examples based on pixel intensity changes on the MNIST `ConvBig` network.

In Figure 6, we show adversarial examples for the MNIST `ConvSmall` network. The 3 sub-figures represent the digits 2, 9, and 9, but the images in our regions are classified as 3, 5, and 4, respectively. Our regions are of size $10^{520}$, $10^{652}$, and $10^{708}$, respectively.

In Figure 7, we show adversarial examples for the MNIST $8 \times 200$ network. The 3 sub-figures represent the digits 5, 9, and 2, but the images in our regions are classified as 8, 8, and 3, respectively. Our regions are of size $10^{148}$, $10^{128}$, and $10^{157}$, respectively.

In Figure 8, we show adversarial examples for the CIFAR10 `ConvSmall` network. The 3 sub-figures represent a boat, a bird, and another bird, but the images in our regions are classified as an airplane , a frog, and a dog, respectively. Our regions are of size $10^{626}$, $10^{563}$, and $10^{608}$, respectively.

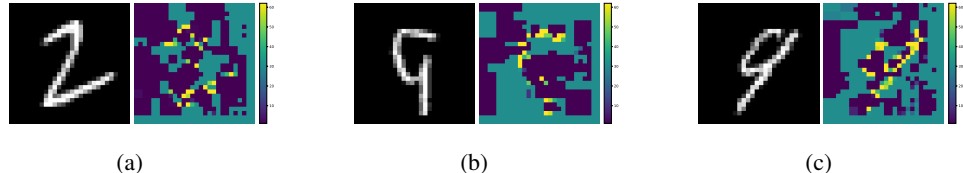

|   (a)   |   (b)   |   (c)   |

Figure 6: Sensitivity of probably robust adversarial examples based on pixel intensity changes on the MNIST `ConvSmall` network.

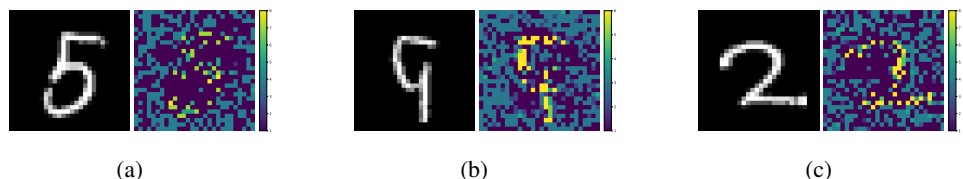

|   (a)   |   (b)   |   (c)   |

Figure 7: Sensitivity of probably robust adversarial examples based on pixel intensity changes on the MNIST $8 \times 200$ network.

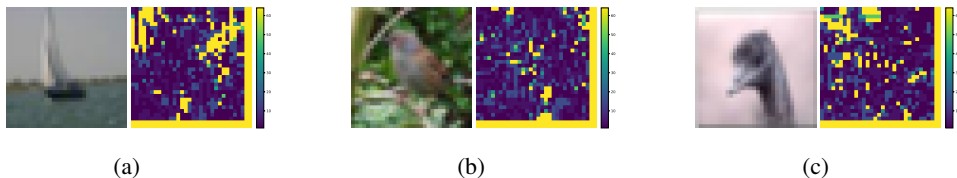

|   (a)   |   (b)   |   (c)   |

Figure 8: Sensitivity of probably robust adversarial examples based on pixel intensity changes on the CIFAR10 `ConvSmall` network.

### F.3 ADVERSARIAL EXAMPLES ROBUST TO GEOMETRIC CHANGES

In Figure 9 – 11, we visualise adversarial examples provably robust to geometric perturbations for the different networks in Table 2. For all figures, the colorbar on the right-hand side quantifies the number of values each pixel can take in our inferred region. The yellow and violet colors represent the two extremes. The intensity of the yellow-colored pixels can vary the most, thus these pixels contribute more to the adversarial examples. For all figures, the subfigures represent an adversarial example for each of the rows in Table 2.

In Figure 9, we show adversarial examples for the MNIST `ConvBig` network. The 3 sub-figures represent the digits 6, 1, and 3, but the images in our regions are classified as 5, 8, and 7, respectively. Our regions have underapproximations of size $10^{159}$, $10^{183}$, and $10^{13}$, respectively.

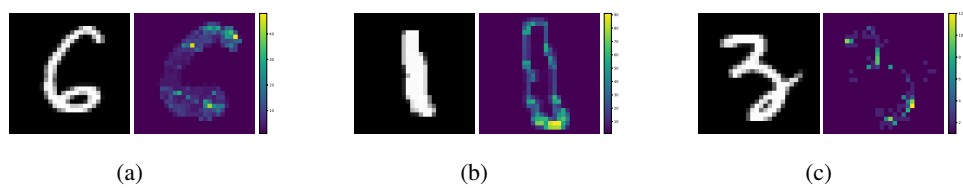

|   (a)   |   (b)   |   (c)   |

Figure 9: Sensitivity of probably robust adversarial examples based on geometric perturbations. Subfigures (a), (b) and (c) correspond to the first, second, and third geometric perturbations for the MNIST `ConvBig` network in Table 2, respectively.

In Figure 10, we show adversarial examples for the MNIST `ConvSmall` network. The 3 sub-figures represent the digits 6, 1, and 3, but the images in our regions are classified as 5, 8, and 7, respectively. Our regions are of size $10^{115}$, $10^{82}$, and $10^{63}$, respectively.

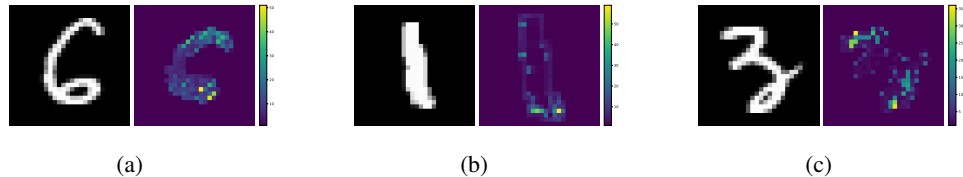

(a)                              (b)                              (c)

Figure 10: Sensitivity of probably robust adversarial examples based on geometric perturbations. Subfigures (a), (b) and (c) correspond to the first, second, and third geometric perturbations for the MNIST `ConvSmall` network in Table 2, respectively.

