# OpenReview forum: "Provably Robust Adversarial Examples"
_ICLR.cc/2022/Conference — ICLR 2022 Poster_

### Official Review · Reviewer_XdXU · 2021-10-30

**Correctness:** 3
**Technical Novelty And Significance:** 2
**Empirical Novelty And Significance:** 3
**Recommendation:** 6
**Confidence:** 4

**Main Review:**

The term "provably robust" appears misleading; there is no theory showing that the examples must be adversarial.

While authors highlight that there are massive amount of adversarial examples (say $10^{573}$) produced by the algorithms, such number seems really dependent on particular problems while lacking a theoretical justification.

On the novelty of the algorithms, I feel it relies many black-box components and their properties; which lowers the technical contribution if the work.


**Updates after discussion**

I agree that the paper brings out interesting ideas and the experimental results are convincing. However, I also feel authors need to tune down the contributions on the theoretical part because many of the guarantees hinge on black-box components that are leveraged from prior works.

**Summary Of The Paper:**

The main contribution of the paper is a framework that will output a large batch of adversarial examples, assuming access to a few black-box mechanisms.

**Summary Of The Review:**

See above.

---

> ### Author Response · Authors · 2021-11-16
> **Response to Reviewer XdXU Part 1/1**
>
> We thank the reviewer for their questions. Below we answer these and hope our answers are sufficient to address all of reviewer's concerns.
>
> **Q:** The term "provably robust" appears misleading; there is no theory showing that the examples must be adversarial.
>
> **A:** A provably robust example is successfully generated only if the used verifier (referred to as verifier \mathds{V} in the paper ) succeeds to prove that it only contains adversarial examples (Line 6 in Algorithm 1) following the definitions in Appendix A.2. We have added Theorem 1 in Appendix C.1 detailing this in the latest version of the paper. Therefore, all provably robust examples in Table 1 and 2 are **guaranteed** to be adversarial. Further, in Appendix C.2 we explore the conditions under which Algorithm 1 is guaranteed to converge to a provable adversarial example, i.e to succeed. In our experiments, we use adversarial attacks that use noise levels small enough that the perturbations do not change the semantics of the images. Please see Appendix E for visualisations of the generated provable adversarial examples.
>
> **Q:** Authors highlight that there are a massive amount of adversarial examples produced by the algorithms. However the number seems really dependent on particular problems while lacking a theoretical justification.
>
> **A:** We agree with the reviewer that the exact number of adversarial examples is related to the particular neural network input domain that is used. However, we believe any network that depends on many input parameters that can vary (e.g pixels in images, audio waveforms in sound or characters in text ) will produce exponentially many individual adversarial instances which cannot be obtained by simply running standard adversarial example techniques.
>
> **Q:** On the novelty of the algorithms, I feel it relies on many black-box components and their properties. This lowers the technical contribution of the work.
>
> **A:** We disagree with the reviewer that our algorithm has a low technical contribution. Despite the use of black box tools, our methods combine them in novel and technically complex ways to arrive at Algorithm 1. Further the extension of our algorithm in Appendix D to polyhedral regions and the theory provided in Appendix C regarding the convergence of Algorithm 1 are also non-trivial. Further, as reviewer A11V points out, the reliance on black box algorithms is a plus of our method, as it allows us to benefit from future advances in neural network verification methods out of the box for efficiency gains and also allows us to easily adapt to new types of adversarial examples/transformations.

---

> > ### Comment · Reviewer_A11V · 2021-11-19
> > **Clarifying the term "provably robust adversarial examples"**
> >
> > To the authors:
> >
> > I think it might be worth clearly explaining why you're referring to an input region containing points guaranteed to be adversarial as a "provably robust adversarial example", since it seems to have been confusing for the reviewer, and was confusing to me when I first read the paper. (In particular, it's not immediately obvious why an input _region_ corresponds to an adversarial _example_, when "example" typically refers to a single input point).
> >
> > As I understand it, one should think of the _center_ of the input region as the "provably robust adversarial example"; this example can be perturbed anywhere within the input region and is still adversarial. If this is the case, perhaps it's worth making that framing clear in the main body of the paper.

---

> > > ### Author Response · Authors · 2021-11-21
> > > **Re: Clarifying the term "provably robust adversarial examples"**
> > >
> > > To Reviewer A11V:
> > >
> > > We agree with the reviewer that our provably robust adversarial examples can also be viewed as regular adversarial examples that come with the additional property that they are enclosed within a provably robust adversarial region calculated by our method. We have modified our definition of provably robust adversarial examples in the introduction section of the newest version of the paper to clarify this.

---

### Official Review · Reviewer_N75e · 2021-10-30

**Correctness:** 3
**Technical Novelty And Significance:** 3
**Empirical Novelty And Significance:** 3
**Recommendation:** 6
**Confidence:** 3

**Main Review:**

### Strength

This paper has clearly stated its objective, the approach and the corresponding evaluations. The proposed technique is designed for a concrete objective, generating provably-robust adversarial examples. The approaches are well-documented in the paper and evaluations are conducted over several datasets and models.
The paper has spent a lot of space explaining the technique from a high-level perspective down to its implementation details, which helps the reader to better understand the algorithm.

### Weakness

My concerns of the paper mainly focus on the following three aspects:

1. Motivation of the “provable” part of the adversarial examples is missing. This paper relates to the prior work in generating robust adversarial examples [1, 2], where they can serve as a good motivation to generate “robust'' adversarial examples”: these papers discuss several physical distortions in applying the adversarial examples into the real-world cases for images and audio. In the other word, these distortions are real-world adversaries for artificial adversarial examples. However, the motivation for the “provable” part is missing to me. I understand the “provable” part can be related to a counter problem, probably-robust networks. The provably-robust network is motivated to build networks where the robustness can be guaranteed for all possible attacks and the evaluations are free from the choice of adversaries. To that end, can the authors explain more about the motivation for “provably”-robust adversarial examples? A follow-up question is: does the robust region proposed in this paper actually contain the physical distortions that may be encountered in the real-word cases [1, 2] and how often? It seems that the more important part we need to prove is that these regions are guaranteed to contain all or part of the distortions you can possibly encounter so that an adversary does not need to worry about that an adversarial example fails in practice.

2. Important experimental setups and discussions are missing.
- Unfortunately with some amount of time during my review I can not locate the concrete definitions and actual implementations of the intensity changes and geometric changes as mentioned in Table 1 and 2. This information should be helpful to understand the importance of the results.

- How “provable” is evaluated? Table 1 and 2 seem to only evaluate how big the region is and Table 3 seems to use randomized smoothing as an attack to adversarial examples generated by the proposed approach. However, the motivation of the paper mostly relies on [1, 2], where the “robustness” of an adversarial example is actually not designed against a prediction defense, i.e. randomized smoothing, but transformations and distortions.

- Unless I misunderstand the results, Table 1 and 2 seem to only aggregate over less than 100 examples per dataset and it may take up to ~5000s for one example in CIFAR. I understand that the bottleneck is that the verifier is usually resource-consuming. If that is the case, the authors may need to convince the readers under what circumstances this trade-off between resource and probably-robustness is worthwhile compared to the fast empirical approaches.

3. Writing. I find the writing of the method part is well-organized and polished, which makes me enjoy the reading of the approach. However, the experiment part is relatively dense and sometimes even difficult to read when a lot of notations and symbols appear in the paragraph without sufficient explanations to remind the reader what they refer to. Also, it would be best to add explanations to notations in the captions of figures and tables so the reader does not have to search for what is measured in the table.

[1] Athalye, A., Engstrom, L., Ilyas, A., & Kwok, K. (2018, July). Synthesizing robust adversarial examples. In International conference on machine learning (pp. 284-293). PMLR.

[2] Qin, Y., Carlini, N., Cottrell, G., Goodfellow, I., & Raffel, C. (2019, May). Imperceptible, robust, and targeted adversarial examples for automatic speech recognition. In International conference on machine learning (pp. 5231-5240). PMLR.


**Summary Of The Paper:**

The manuscript introduces a definition of provablely-robust adversarial examples, a set of examples that are verified to be classified as different labels compared with the input of interest. The main idea of the technique is to shrink a box-like region from an over approximation to a verifiable smaller sub-region such that a robustness verifier will return robust for all points in that particular sub-region. In the evaluation part, the author demonstrates the effectiveness of the approach with several experiments, i.e. robustness against intensity transformation and randomized smoothing defense.


**Summary Of The Review:**

Overall I incline to a weak rejection at this stage of the reviewing process but I am open to any discussions. The reasons that prevent me from giving higher scores are the insufficient descriptions of the motivations and the current way the experiment sections are written with, which I have mentioned in my main review.

---

> ### Author Response · Authors · 2021-11-16
> **Response to Reviewer N75e Part 1/2**
>
> We thank the reviewer for the detailed review. We now proceed to answer the questions. We hope our answers are sufficient to address all of the reviewer's concerns.
>
> **Q:** Motivation of the “provable” part of the adversarial examples is missing.
>
> **A:** The primary motivation is to take existing attack methods that generate individual attacks and make them stronger by generating provable regions. Our smoothing experiments in Table 3 demonstrate that our attacks indeed achieve their primary goal of making the original attack methods stronger since taking the center of our regions results in stronger attacks. We believe one can use these stronger attacks to better evaluate the adversarial robustness of networks compared to empirically robust attacks. We show in Section 5.5, that our provable examples are more reliable for assessing the robustness of networks to adversarial examples than empirical ones as the latter misses many examples close to the original image giving a false sense of security. Finally, we point out that the pixel intensity and the geometric transformations considered in Section 5.1 and 5.2 are a subset of the physical distortions that can occur in real-world scenarios. We believe that pairing our methods with a wider range of attacks, including different types of physical distortions, many of which are not yet supported by current verification tools, is an important future direction of research.
>
> **Q:** How “provable” is evaluated? Table 1 and 2 seem to only evaluate how big the region is and Table 3 seems to use randomized smoothing as an attack to adversarial examples generated by the proposed approach. However, the motivation of the paper mostly relies on [1, 2], where the “robustness” of an adversarial example is actually not designed against a prediction defense, i.e. randomized smoothing, but transformations and distortions.
>
> **A:** We show the sizes of our robust examples in Table 1 and 2, in order to provide a measure of their robustness. In Table 3, we demonstrate that the robustness of our examples can be leveraged to attack a smoothed classifiers $g$, by attacking them with the concrete midpoint of our robust adversarial regions, which is a point in a region with a high concentration of adversarial examples, guaranteed by the provability of our adversarial regions. We evaluate the effectiveness of these midpoints based on the maximal verifiable radius $R_{adv}$ on the smoothed classifier $g$.
>
> **Q:** Important experimental setups and discussions are missing. The experiment part is relatively dense and sometimes difficult to read when a lot of notations and symbols appear in the paragraph without sufficient explanations to remind the reader what they refer to.
>
> **A:** We have provided more experimental setup details for Sections 5.1/5.2/5.3/5.4, as well as added more informative captions to Table 1 and 2. Further, we made changes to Section 5.4 (previously 5.3) in order to better explain our method of assessing the strength of our adversarial examples against smoothing. We hope this will help the reviewer understand the experimental setup much better. If parts of it are still unclear, we request the reviewer to list concrete missing/unclear points and we will happily address them in the main body or the appendix of the paper depending on the available space.
>
> **Q:** I can not locate the concrete definitions and actual implementations of the intensity changes and geometric changes as mentioned in Table 1 and 2.
>
> **A:** We have supplied precise definitions in Appendix A.2 in the new version of the paper.

---

> ### Author Response · Authors · 2021-11-16
> **Response to Reviewer N75e Part 2/2**
>
> **Q:** Table 1 and 2 seem to only aggregate over less than 100 examples per dataset and it may take up to ~5000s for one example in CIFAR. I understand that the bottleneck is that the verifier is usually resource-consuming. The authors may need to convince the readers under what circumstances this trade-off between resource and probably-robustness is worthwhile compared to the fast empirical approaches.
>
> **A:** Evaluating on the first 100 test images is a standard practice in the verification literature - e.g. see [1], [2] and [3]. We note that in order to scale our method further, we can take advantage of the recent GPU implementation of DeepPoly (see [4]) that takes advantage of the huge parallelism GPUs provide. As discussed above, our provable examples allow us to better assess the overall robustness of the network to adversaries compared to empirical approaches due to the guarantees provided by them.
>
> **Q:** Add explanations to notations in the captions of figures and tables so the reader does not have to search for what is measured in the table.
>
> **A:** We have improved section 5.3 and added additional details to Sections 5.1 and 5.2 in the latest version of the paper. Further we added better captions to Table 1 and 2.
>
>
> **[1]** Mislav Balunovic, Maximilian Baader, Gagandeep Singh, Timon Gehr, and Martin Vechev. Certifying ´ geometric robustness of neural networks. Advances in Neural Information Processing Systems 32, 2019.
>
> **[2]** Gagandeep Singh, Timon Gehr, Matthew Mirman, Markus Püschel, and Martin Vechev. Fast and effective robustness certification. In Advances in Neural Information Processing Systems, pp. 10802–10813, 2018a.
>
> **[3]** Gagandeep Singh, Timon Gehr, Markus Püschel, and Martin Vechev. An abstract domain for certifying neural networks. Proceedings of the ACM on Programming Languages, 3(POPL):41, 2019.
>
> **[4]** Müller, Christoph, et al. "Scaling Polyhedral Neural Network Verification on GPUs." arXiv preprint arXiv:2007.10868 (2020).

---

> > ### Comment · Reviewer_N75e · 2021-11-21
> > **Response to Authors**
> >
> > Thank you for the responses and the revision. I believe that the authors have managed to resolve my most of my initial concerns in the evaluation part. Even though the motivation and the significance of the paper still remain a bit weak to me (as the current evaluation uses randomized smoothing as an "adversary" instead of a super set of transformations, which may block an adversary in practice, of intensity changes and geometric changes considered in this paper), I would like to increase the score to reflect the authors' effort on addressing my other concerns.
> >
> > > Finally, we point out that the pixel intensity and the geometric transformations considered in Section 5.1 and 5.2 are a subset of the physical distortions that can occur in real-world scenarios.
> >
> > A minor suggestion: as several reviewers have mentioned that the term "provably robust" is bit misleading when it is used with "adversarial example" at the same time. I keep reminding myself that the robustness discussed in this paper refers to that the adversarial examples are still misclassified instead of another situation that the network is able to correctly classify these examples (the previous work [1] uses "robust attack" instead of "robust adversarial example", which seems to be less confusing). It would be appreciated if the authors plan to clarify the term in the paper or replacing the term "robust adversarial example" with some other terms, e.g. "valid adversarial example", in the future (you don't have to update the paper now).
> >
> > [1] Mislav Balunovic, Maximilian Baader, Gagandeep Singh, Timon Gehr, and Martin Vechev. Certifying ´ geometric robustness of neural networks. Advances in Neural Information Processing Systems 32, 2019.

---

> > > ### Author Response · Authors · 2021-11-22
> > > **Further Response to Reviewer N75e Part 2/2**
> > >
> > > We thank the reviewer for the helpful comments on the paper. We will make the proposed change of terminology in the next revision of the paper.

---

### Official Review · Reviewer_A11V · 2021-10-31

**Correctness:** 4
**Technical Novelty And Significance:** 4
**Empirical Novelty And Significance:** 3
**Recommendation:** 8
**Confidence:** 5

**Main Review:**

# Strengths

- **Originality**: The paper presents a novel algorithm to generating provably robust adversarial examples corresponding to semantically meaningful regions.
- **Quality**:
  - The paper demonstrates that the algorithm can scale to generate provably robust adversarial examples of non-trivial size on networks of non-trivial size.
  - The experiments in the paper have clearly been carried out with an attention to detail.
- **Clarity**:
  - The paper describes the algorithm in sufficient detail to enable reproducibility. (In particular, the appendix explains important details that would be required to re-implement the approach.)
- **Significance**:
  - The approach presented is modular, using existing certification algorithms as a subroutine. This has two key benefits:
    - Improvements to existing certification algorithms can be used to improve the search efficiency for provably robust adversarial examples
    - Certifiers which handle new classes of transformations could be used to generate provably robust adversarial examples for these classes of transformations.
  - While this paper focuses on adversarial examples, the approach can be used in any setting where we are interested to find large regions of input space with a constant classification (or, more generally, where a linear function of some neuron activations exceeds a threshold). I can imagine this being applied to better understanding/visualizing how the output of a neural network varies as the inputs change.

# Areas for Improvement

## Originality

- In the introduction, the paper states that "our regions are guaranteed to be adversarial while prior approaches are empirical and offer no such guarantees". Section B.2. mentions Liu et al., which "is also capable of generating provably robust hyperbox regions". Is the statement in the introduction wrong?

## Quality

- The baseline used seems to be a straw man, since it is simply "our method" but with uniform rather than adaptive shrinking; I would always expect "our method" to outperform the baseline. I would prefer to see the comparison to Liu et al. (and any other methods that produce provably adversarial regions if they exist) in the main body of the paper instead.
- In Table 2, the transforms selected appear quite arbitrary; in particular, they appear like they could have been cherry-picked to flatter the presented approach. Some detail on how the transformations were selected would alleviate this concern.

## Clarity

- Experimental setup for Section 5.3:
  - I struggled to understand what experiment was run in this section and what the results in Table 3 show. I understand that the goal of this section is to show that robust adversarial examples are significantly more effective against defenses based on randomized smoothing, but the setup for the experiment is still not clear to me. I'd be happy to discuss this with the authors, but some preliminary questions:
    - What are the units of 'robustness'?
    - Is the result for "Ours" normalized to 1?
    - Table 3:
      - Are the results for "baseline" and "ours" mean, or some other summary statistic?
      - What result exactly is shown for "individual attacks"? Were multiple attacks generated for each image, or was the individual attack the attack that was used to determine the value of $R'$?
- Reporting # of concrete examples:
  - In Table 1, the SizeO column reports an _upper bound_ on the number of concrete examples in the polyhedral region. This is not immediately clear from the description; a reasonable reader might expect that this is just the number of concrete examples. I would request that the authors either:
    - Estimate the actual number of concrete examples
    - Clearly indicate in the table description that this is an overestimate
    - Remove the SizeO column.
  - I have a similar concern with the "Over" column in Table 2; I don't see how an overestimate of the number of concrete examples in the region is relevant.

# Additional Comments

## Clarity

Here are some issues with clarity that do not constitute major issues, but would still improve the paper significantly if addressed. At the high level, he paper appears to be trying to squeeze too many results in, leading to important details being omitted.

### Missing Details

- Section 5.3: "We exclude images whose radius is too big" - what constitutes too big? For these images, what is the robustness to $L_2$ smoothing defenses of adversarial examples generated by your method?
- Table 1 / Table 2: Is the time reported here a median or average, and is it only for instances where the methods succeed?
- Table 2: The value of #Splits is listed but no guidance is provided to the reader as to how to interpret the result. I would recommend moving this information to the appendix or adding an interpretation.
- Definition 2: "whose $L_2$ ball is certified as adversarial" - I didn't find a definition in the paper of what it means for the $L_2$ ball to be adversarial. (I would have assumed that this means that every sample in the ball has a different classification as compared to $x$, and not that every sample has to have the same classification as $\tilde{x}$, but the rest of the paper seems to suggest the latter definition.)

### Miscellaneous Points

- Section 3 (Overview): "... assumes an algorithm $\alpha$" - the variable $\alpha$ is already used above to indicate the probability that `CERTIFY` fails. I'd recommend using a different variable here.
- Section 3.2 (Computing an Underapproximating Region): "sacrificing a few pixels where the network is not very robust" - did you mean where the _adversarial example_ is not very robust here? If the network is not robust for a certain pixel, it doesn't make sense to me to sacrifice _those_ pixels ...
- Section 5.1: "Column #Regions" is referenced, but it is "#Reg" in both tables.

## Spelling / Grammar

- Section 2.2 (Geometric Certification): "creates overapproximation of the region" -> "creates an approximation of the region"
- Figure 1: "repred crosses" -> "red crosses"?

# Questions

This is out of the scope of this paper, but the result in Section 5.4 suggests that it might be possible to find perturbations to empirically robust adversarial examples (empirically verified by an EoT approach) that result in a correctly classified image. Do you have any sense whether it would be possible to consistently find such "dis-adversarial attacks" on empirically robust adversarial examples?

**Summary Of The Paper:**

This paper presents a novel algorithm for identifying "provably robust adversarial examples": large regions in the input space that provably contain only adversarial examples.

Each region corresponds to a single adversarial example $\tilde{x}$ found in the center of the region, along with all the points that can be generated by applying a sequence of transformations to $\tilde{x}$. The transformations considered in the paper are semantically meaningful changes to the original image. Critically, we can be guaranteed that $\tilde{x}$ will be misclassified even if _it_ is perturbed.

The paper demonstrates that the algorithm can generate regions of non-trivial size on networks of non-trivial size. For example, for a CIFAR10 classifier with 6 layers and ~62k neurons, it finds axis-aligned regions containing a median of $10^{573}$ adversarial examples. In addition, the paper shows that provably robust adversarial examples can be used to create adversarial examples to $L_2$-smoothed classifiers that are more robust to $L_2$ noise as compared to adversarial examples generated directly via PGD attacks.

**Summary Of The Review:**

Overall, I recommend accepting the paper. The paper presents a novel approach to finding large regions of adversarial examples, with strong experimental evidence that it scales well. The details provided would enable other researchers to reproduce the presented approach. Most importantly, this approach is likely to be something that other researchers can use and build upon.

Having said that, the paper has some issues with clarity. Details are provided in the main review, but I'd like to highlight in particular Section 5.3, which I found particularly hard to parse.

N.B. My current recommendation for this paper as-is is 6, but I'd be quite happy to upgrade the recommendation to 8 if the bulk of my concerns around clarity are addressed.

---

## After Paper Discussion Period
During the paper discussion, the authors addressed the bulk of my concerns around clarity, and I've upgraded my recommendation to 8 as a result.

---

> ### Author Response · Authors · 2021-11-16
> **Response to Reviewer A11V  Part 1/3**
>
> We thank the reviewer for the exhaustive review and questions. Below we answer all of their questions. We hope our answers are sufficient to address all of the reviewer's concerns.
>
> **Q:** In the introduction, the paper states that "our regions are guaranteed to be adversarial while prior approaches are empirical and offer no such guarantees". Section B.2. mentions Liu et al., which "is also capable of generating provably robust hyperbox regions". Is the statement in the introduction wrong?
>
> **A:** The statement in the introduction is referring specifically to the concept of provably robust adversarial examples, which to the best of our knowledge, is introduced here for the first time. Hence, ours is also the first work to propose algorithms for generating such provable adversarial examples. The statement in Section B.2 (now Section 5.3) is referring to the fact that Liu et al. can be (partially) extended (see also answer to next question) for solving the problem of generating provably robust adversarial examples. We have adjusted the claims in Section 5.3 accordingly.
>
> **Q:** The baseline used seems to be weak, since it is simply "our method" but with uniform rather than adaptive shrinking; I would prefer to see the comparison to Liu et al. (and any other methods that produce provably adversarial regions if they exist) in the main body instead.
>
> **A:** To the best of our knowledge,  Liu et. al is the only work that can be adapted to a subset of our benchmarks. It does not generate regions for geometric perturbations or convolutional networks and only produces hyperbox regions for fully-connected networks. Therefore we can only run it on our 8x200 fully-connected network for generating adversarial examples robust against intensity changes. Our results show that it performs significantly worse than our method. We have now brought the comparison with Liu et al. into the main paper as Section 5.3 in the revised version. As a result of the limitation of the adaptation of Liu et al. we consider comparison with uniform shrinking relevant as it is the only baseline that works in all of our settings and benchmarks.
>
> **Q:** In Table 2, the transforms selected appear quite arbitrary
>
> **A:** Adversarial examples based on geometric transformations with only a single parameter can be trivially obtained with brute-force search. Therefore, we focus on experiments with 3 or 4 parameters to make the setting more challenging and reflect the perturbations considered in the state-of-the-art verifier for geometric perturbations [1]. We have chosen the parameter combinations so that all transformations from [1] are present (Rotation, Scaling, Sheering, Brightness, Translation). We chose the search range for each parameter so that the attackable number of images are both not too small and not too large, similarly to how we have chosen the $\epsilon$ in Table 1. We now added additional text in the revision clarifying this point in Section 5.2.
>
> **Q:** I struggled to understand what experiment was run in Section 5.3 and what the results in Table 3 show. The setup for the experiment is still not clear to me.
>
> **A:** In the revised paper, Section 5.3 was rewritten and moved to 5.4. Further, Definitions 1 and 2 in Section 2.3 were modified to make them simpler to follow. Further, we have moved some of the details about the experimental setup to Appendix A.6, including details about the neural networks used and an expanded explanation on how $R’$, $\sigma$ and $\alpha$ from Definition 2 are chosen. We used the additional space in the main body to provide important clarifications, including better explanation of what the numbers in Table 3 are and how they are computed and what the individual attacks are and how they are computed. We further provided a better discussion of the results observed in Table 3. We hope this helps the reviewer to understand the experimental setup better. If parts of it are still unclear, please specify which and we will clarify them further.
>
> **Q:** Experimental setup for Section 5.3. What are the units of 'robustness'? Is the result for "Ours" normalized to 1?
>
> **A:** The units of robustness that we show in Table 3 represent the ratio between the adversarial strength $R_{adv}$ (Definition 2) calculated for the other methods and our method. Since Table 3 depicts ratios, the measurement is relative and the row depicting our method is fixed to 1. The adversarial strength $R_{adv}$ is defined in terms of the largest certifiable adversarial radius around the attack $\tilde{x}$. We have clarified these points in Section 5.4 in the revised paper.
>
> **Q:** Are the results for "baseline" and "ours" mean, or some other summary statistic?
>
> **A:** For all 4 methods in Table 3 we calculate a mean across the first 100 images in the respective test sets. We added a clarification in Section 5.4 of the revised paper.

---

> > ### Comment · Reviewer_A11V · 2021-11-19
> > **Response for initial author response**
> >
> > - Thanks for bringing the comparison with Liu et al. into the main body of the paper.
> > - I also appreciate the explanation on why certain transformations were selected, since that makes the comparison seem less arbitrary.
> > - The setup for the experiment that is being run in Section 5.4 (previously section 5.3) is now clearer to me, thank you. It seems obvious to me that the adversarial examples generated by your method (and the baseline) should have a larger value of $R_adv$ on the original network $f$ as compared to randomly selected adversarial attacks, since your method optimize over $R_adv$ while randomly selected adversarial attacks do not. Is the point here that your method _also_ generated adversarial examples with a large value of $R_adv$ on the smoothed network $g$? If so, that point should be made clearer. The first sentence of Section 5.4 says instead that "our adversarial examples robust to intensity changes are significantly more effective against state-of-the-art defenses based on randomized smoothing" -- if that's what we want to show in this section, why not compute a simple statistic like the proportion of successful attacks on $g$, rather than a proxy like $R_adv$?

---

> > > ### Author Response · Authors · 2021-11-21
> > > **Further Response to Reviewer A11V Part 1/3**
> > >
> > > We are glad the reviewer found the changes we made adequate and the description of the experimental setup in Section 5.4 clearer. Below we answer the outstanding questions:
> > >
> > > **Q:** It seems obvious to me that the adversarial examples generated by your method (and the baseline) should have a larger value of $R_{adv}$ on the original network $f$ as compared to randomly selected adversarial attacks, since your method optimize over $R_{adv}$ while randomly selected adversarial attacks do not.
> > >
> > > **A:** The results of the experiment in Section 5.4 confirm our claim that our method can be used to strengthen traditional adversarial attack methods like the PGD, which our individual attacks are based on, in order to obtain examples with robustness unachievable by sampling from PGD alone. Further, we want to clarify that our adversarial regions do not directly optimize $R_{adv}$, as $R_{adv}$ is calculated on $g$, while we only maximize our provable adversarial regions’ size on $f$. To this end, the experiment we present in Section 5.4 is important to demonstrate that the robustness of our regions on the undefended network $f$ transfers to $g$, which is harder to attack compared to $f$, as it is defended using smoothing.
> > >
> > > **Q:** Is the point here that your method also generated adversarial examples with a large value of $R_{adv}$ on the smoothed network $g$? If so, that point should be made clearer. The first sentence of Section 5.4 says instead that "our adversarial examples robust to intensity changes are significantly more effective against state-of-the-art defenses based on randomized smoothing" -- if that's what we want to show in this section, why not compute a simple statistic like the proportion of successful attacks on $g$, rather than a proxy like $R_{adv}$?
> > >
> > > **A:** As we explained in the previous question, $R_{adv}$ is indeed calculated on the smoothed network $g$. We have further clarified this point in Section 5.4 and Definition 2 in the newest version of the paper. One of the goals of Section 5.4 is to demonstrate that our method is capable of robustifying the underlying attack method \mathds{A}, which in Section 5.4 is the standard $L_2$ PGD attack on the smoothed classifier $g$. A natural way to show this is to compute the certified radius $R_{adv}$, which is already used in smoothing as a measure of robustness of individual images. Further, the certified radius $R_{adv}$ also measures the size of our robust adversarial regions on $g$ and, thus, allows us to show that the large sizes of our regions computed on $f$ and shown in Section 5.1 and 5.2 translate to large certified radii $R_{adv}$ on $g$.
> > >
> > > **[1]** Jeremy Cohen, Elan Rosenfeld, and Zico Kolter. Certified adversarial robustness via randomized smoothing. In International Conference on Machine Learning, pp. 1310–1320. PMLR, 2019.

---

> > > > ### Comment · Reviewer_A11V · 2021-11-21
> > > > **Response to authors' further response**
> > > >
> > > > Thanks for your update; I hope the changes not only clarify the contributions of the paper for me but also for future readers.
> > > >
> > > > Considering the discussion period as a whole, I've updated my recommendation for the paper to 8.

---

> ### Author Response · Authors · 2021-11-16
> **Response to Reviewer A11V Part 2/3**
>
>
> **Q:** What result exactly is shown for "individual attacks"?
>
> **A:** For all individual attacks generated by the adversarial algorithm \mathds{A} (previously $\alpha$), we apply Definition 2 to obtain the adversarial strength measure $R_{adv}$. Table 3 depicts the ratio between the other methods’ adversarial strength and the adversarial strengths of our method.  Table 3 depicts the mean and 95-percentile strength ratio values calculated across all the attacks generated by \mathds{A}. We have clarified this in Section 5.4 in the revised paper.
>
> **Q:** Were multiple attacks generated for each image, or was the individual attack the attack that was used to determine the value of $R’$?
>
> **A:** The same 500 individual attacks from the adversarial algorithm \mathds{A} (previously $\alpha$) were used to select $R’$, generate the robust adversarial examples and were evaluated in the table under “Individual attacks”. We have clarified this point in Appendix A.6 in the revised paper.
>
> **Q:** Section 5.3: "We exclude images whose radius is too big" - what constitutes too big? For these images, what is the robustness to smoothing defenses of adversarial examples generated by your method?
>
> **A:** We have excluded images whose adversarial example radius $R'$ is more than 33% bigger than the certified radius $R$ of the original image $x$. We chose the 33% empirically, as we found that for images with bigger values for $R'$  almost all classes are adversarial, suggesting that finding robust adversarial regions for $g$ is trivial. We have clarified that in Appendix A.6 in the latest version. For the excluded images in the smoothing experiment in Table 3, we report the strength ratios relative to our method and therefore our method’s table row would stay at 1.0. We expect that the attack strength ratio for the other methods will be closer to 1.0, i.e closer to our method, as the attacks will be much easier to execute.
>
> **Q:** SizeO and Over columns in Table 1 and 2 report an upper bound on the number of concrete examples. This is not immediately clear from the description. I don't see how an overestimate of the number of concrete examples in the region is relevant.
>
> **A:** Estimating the actual number of concrete points inside a high dimensional polyhedra is in general an intractable problem (e.g. see [2]). We have adjusted Tables 1 and 2 to better indicate that the resulting numbers are an overapproximation. We believe that the inclusion of these numbers is helpful to the reader, as they provide an upper bound on the largest possible region found by our method. Further, it can be the case that the overapproximation region can actually be verified by a more precise but more expensive (complete) verifier.
>
> **Q:** Table 1 / Table 2: Is the time reported here a median or average, and is it only for instances where the methods succeed?
>
> **A:** The time reported is the average time on instances for which at least one adversarial attack is found by the adversarial algorithm. We have clarified this in Sections 5.1 / 5.2 in the latest version of the paper.
>
> **Q:** Table 2: The value of #Splits are listed but no guidance is provided to the reader as to how to interpret the result.
>
> **A:** The cost and the precision of our method increases with the number of splits. Therefore, to allow for a fair comparison we chose the number of splits, so that the baseline runs in similar or higher runtime compared to our method. This setting is slightly favouring the baseline. We have clarified this point in Section 5.2 in the revised version. We think the inclusion of the number of splits and this explanation in the new version of the paper is important for the reader to better understand our experimental setup.
>
> **Q:** I didn't find a definition in the paper of what it means for the L2  ball to be adversarial. The paper seems to suggest that every sample has to have the same classification as
> $\tilde{x}$, while I expect that every sample in the ball has a different classification as compared to $x$.
>
> **A:** We confirm we use the “same classification as $\tilde{x}$” definition, as it allows us to directly apply the Certify procedure from [3]. We note that in practice we do not believe this distinction is important for the results in Section 5.4, as for regions close to the original $x$ (as ensured by our choice of $R'$) we find adversarial regions to contain points only from $\tilde{x}$ class and be surrounded by points from only the non-adversarial class. We have clarified this point in Definition 2 in the latest version of the paper.
>
> **Q:** Miscellaneous Points / Spelling / Grammar
>
> **A:** We thank the reviewer for spotting these. We have addressed the problems in the latest version of the paper.

---

> > ### Comment · Reviewer_A11V · 2021-11-19
> > **Response for initial author response**
> >
> > > We have adjusted Tables 1 and 2 to better indicate that the resulting numbers are an overapproximation.
> >
> > Thanks, this is helpful; the inequality symbols provide a useful visual cue that will hopefully prevent readers from making the wrong assumption.
> >
> > > Further, it can be the case that the overapproximation region can actually be verified by a more precise but more expensive (complete) verifier.
> >
> > As far as I understand it, the overapproximation you are using for the polyhedral region is a hyperbox. Do you mean that this hyperbox could be verified with a more expensive complete verifier?
> >
> > ---
> >
> > Finally: I don't mean to belabor the point since the results shown are fine even with only an overapproximation for the number of points within the polyhedral region, but I'm curious about the following points:
> >   - Would computing the number of provably-robust values in the polyhedral region be less computationally intensive than trying to verify the overapproximation hyperbox?
> >   - Can you use a different underapproximation region (e.g. an ellipse) where it's easier to compute the number of points contained within the underapproximation.

---

> > > ### Author Response · Authors · 2021-11-21
> > > **Further Response to Reviewer A11V Part 2/3**
> > >
> > > We are glad that the reviewer found the visual cues added in Tables 1 and 2 helpful. We answer the rest of the questions below:
> > >
> > > **Q:** Can you clarify the statement: “Further, it can be the case that the overapproximation region can actually be verified by a more precise but more expensive (complete) verifier.” ? As far as I understand it, the overapproximation you are using for the polyhedral region is a hyperbox. Do you mean that this hyperbox could be verified with a more expensive complete verifier?
> > >
> > > **A:** We use DeepPoly to verify the polyhedral region. DeepPoly is an incomplete verifier and, therefore, loses precision. What we meant to say with our comment was that because the polyhedral region is verified with DeepPoly, a complete verifier, which is more precise, might be able to verify a bigger region around the polyhedra. It is, therefore, **possible in theory** that the overapproximation hyperbox can be verified by a complete verifier. We point out, however, that in Section 5.1 we experiment on realistic networks to which complete verifiers do not scale. In particular, we use moderately-sized normally trained networks, as opposed to adversarially trained networks which are easier to verify. For these networks the complete verifiers need to explore overwhelmingly many branches and, thus, cannot be applied in reasonable time.
> > >
> > > **Q:** Would computing the number of provably-robust values in the polyhedral region be less computationally intensive than trying to verify the overapproximation hyperbox?
> > >
> > > **A:** They are both NP-complete problems in general. We think that in practice the verification problem might be easier due to recent advancements in complete verification, but we do not believe either computation is practical for the realistic networks we rely on in this work, as explained in the previous question.
> > >
> > > **Q:** Can you use a different underapproximation region (e.g. an ellipse) where it is easier to compute the number of points contained within the underapproximation.
> > >
> > > **A:** Calculating the number of concrete examples in the ellipsoid setting is equivalent to the problem of computing the number of lattice points in an $n$-dimensional sphere. This is a well studied problem and a number of approximations to the exact answer exist (e.g. see [1]) that can result in more precise approximations in reasonable computational time. However, in this work we focus on polyhedral regions, as they lose less precision when being verified using DeepPoly. Using different verifiers that are able to verify ellipsoid regions with smaller precision loss, which we can easily incorporate in our framework, is an interesting future work item.
> > >
> > > **[1]** Henryk Iwaniec and Emmanuel Kowalski. Analytic Number Theory (Colloquium Publications, Vol. 53) (Colloquium Publications (Amer Mathematical Soc)). American Mathematical Society, hardcover edition, 6 2004. ISBN 978-082183633

---

> ### Author Response · Authors · 2021-11-16
> **Response to Reviewer A11V Part 3/3**
>
> **Q:** Section 3.2: "sacrificing a few pixels where the network is not very robust" - did you mean where the adversarial example is not very robust here?
>
> **A:** Yes, correct. We have clarified this in Section 3.2 in the latest version of the paper.
>
> **Q:** The result in Section 5.4 suggests that it might be possible to find perturbations to empirically robust adversarial examples (empirically verified by an EoT approach) that result in a correctly classified image. Do you have any sense whether it would be possible to consistently find such "dis-adversarial attacks" on empirically robust adversarial examples?
>
> **A:** As we point out in Section 5.4 (now 5.5), we find that such subregions tend to be closer to the original trainset/testset image $x$. We think designing algorithms to search for such regions is an interesting item for future work.
>
> **[1]** Mislav Balunovic, Maximilian Baader, Gagandeep Singh, Timon Gehr, and Martin Vechev. Certifying ´ geometric robustness of neural networks. Advances in Neural Information Processing Systems 32, 2019
>
> **[2]** Khachiyan, Leonid. "Complexity of polytope volume computation." New trends in discrete and computational geometry. Springer, Berlin, Heidelberg, 1993. 91-101.
>
> **[3]** Jeremy Cohen, Elan Rosenfeld, and Zico Kolter. Certified adversarial robustness via randomized smoothing. In International Conference on Machine Learning, pp. 1310–1320. PMLR, 2019.

---

### Decision · Program_Chairs · 2022-01-20

**Decision:**

Accept (Poster)

**Comment:**

In this paper, authors introduce and study provably robust adversarial examples. Reviewers had mixed thoughts on the work. One reviewer mentioned that the "provable" robustness is somehow overstated in the work: looking at the title and abstract, it sounds like the paper develops a new algorithm that is guaranteed to be robust, but in reality the robustness hinges on the black-box verifiers (which is acknowledged by the authors during discussion). I agree with this. This should be more clearly stated in the work. I strongly suggest authors to calibrate exaggerated statements of contributions in the revised draft. Having said this, reviewers liked the the experimental study of the paper and found it to be comprehensive and convincing.